# Discrete Dictionary-based Decomposition Layer for Structured Representation Learning

**Taewon Park**[1]     **Hyun-Chul Kim**[1]     **Minho Lee**[1,2]

[1]Kyungpook National University, South Korea
[2]ALI Co., Ltd., South Korea
`ptw7998@gmail.com, hyunchul_kim@knu.ac.kr, mholee@gmail.com`

## Abstract

Neuro-symbolic neural networks have been extensively studied to integrate symbolic operations with neural networks, thereby improving systematic generalization. Specifically, Tensor Product Representation (TPR) framework enables neural networks to perform differentiable symbolic operations by encoding the symbolic structure of data within vector spaces. However, TPR-based neural networks often struggle to decompose unseen data into structured TPR representations, undermining their symbolic operations. To address this decomposition problem, we propose a **D**iscrete **D**ictionary-based **D**ecomposition (D3) layer designed to enhance the decomposition capabilities of TPR-based models. D3 employs discrete, learnable key-value dictionaries trained to capture symbolic features essential for decomposition operations. It leverages the prior knowledge acquired during training to generate structured TPR representations by mapping input data to pre-learned discrete features within these dictionaries. D3 is a straightforward drop-in layer that can be seamlessly integrated into any TPR-based model without modifications. Our experimental results demonstrate that D3 significantly improves the systematic generalization of various TPR-based models while requiring fewer additional parameters. Notably, D3 outperforms baseline models on the synthetic task that demands the systematic decomposition of unseen combinatorial data.[1]

## 1 Introduction

Compositional generalization, aiming at understanding unseen data by combining known concepts, is essential for neural networks to handle complex tasks [2, 13, 12, 16, 8, 6]. Tensor Product Representation (TPR) framework [33] facilitates this by embedding the symbolic structure of data within vector spaces, providing neural networks with compositional capabilities. Within this framework, individual objects are decomposed at the representation level into distinct symbolic components called *role-filler* pairs[2]. The TPR framework encodes each object by taking a tensor product of its *role* vector and *filler* vector, represented as $T = filler \otimes role$, and then superimposes them to represent multiple objects within a single representation. During decoding, the TPR framework retrieves specific *fillers*—essential for solving tasks—from the superimposed representation through matrix multiplication using an *unbinding operator* correlated to a particular *role*, $filler = T \cdot unbind$. This retrieved *filler* is then utilized in downstream tasks. Based on this property, TPR-based neural networks have demonstrated significant generalization and applicability in fields such as associative reasoning [28, 30], mathematical problem-solving [29], and natural language processing [9, 32, 21, 34].

---

[1]The code of D3 is publicly available at https://github.com/taewonpark/D3

[2]The *roles* and *fillers* depend on the task at hand. For example, in a tree structure, the *role* corresponds to a position within the tree, while the *filler* represents the label associated with that position [34]. In associative memory, the *role* is analogous to an associative key, and the *filler* corresponds to the associative value [28, 30].

38th Conference on Neural Information Processing Systems (NeurIPS 2024).

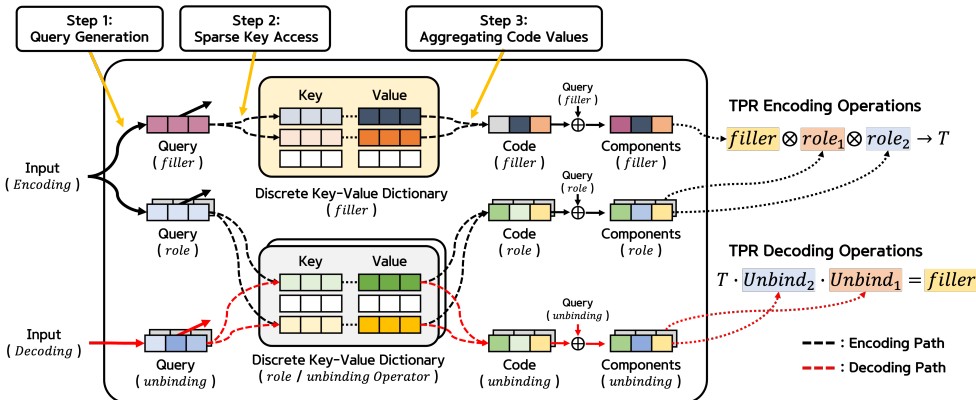

Figure 1: **Overview of D3.** D3 generates structured TPR representations by mapping input data to the nearest pre-learned symbolic features stored within discrete, learnable dictionaries. Each dictionary is linked explicitly to specific TPR components, such as *roles*, *filler*, and *unbinding operators*. Notably, D3 uses a shared dictionary configuration between the *roles* and *unbinding operators*. This figure illustrates, for example, that $role_1$ and $unbind_1$ share one dictionary, while $role_2$ and $unbind_2$ share another. $T$ denotes a superimposed representation that represents multiple objects.

Despite their successes, the TPR-based approaches pose a significant challenge known as a decomposition problem [33, 23], which refers to the difficulty of decomposing input data into TPR components, such as *roles*, *fillers*, and *unbinding operators*. Without accurate decomposition, TPR-based models fail to represent the symbolic structure of data, causing a decline in the performance of the TPR operations. Recently, inspired by an object-centric learning method [18], Park et al. [23] proposes an attention-based iterative decomposition (AID) module to address this issue. AID uses competitive attention to iteratively refine structured representations, thereby enhancing the systematic generalization of TPR-based models. However, it still struggles to generalize all possible combinations of known symbols in simple synthetic tasks. This failure is likely attributable to its insufficient mechanism for explicitly mapping input data to known symbolic features observed during training. Therefore, the decomposition module may need an additional mechanism to store observed symbolic features during training and utilize it to effectively decompose unseen combinatorial data of known symbols.

In another line of work, discrete representation learning has been explored to improve the efficiency, interpretability, and generalization capabilities of neural networks [39, 14, 17, 37, 7]. This approach involves mapping continuous input data into discrete representations by finding the nearest features in a predefined codebook. The features within the codebook are learnable parameters, specifically trained to capture the latent features of data during training phase [39]. Some researchers have applied discrete representation techniques to extract specific types of representations from unstructured data [11, 43, 44]. Other researchers have integrated discrete symbolic embeddings within the TPR framework to improve its interpretability [21, 9]. However, these methods are designed for specific applications, such as question-answering and summarization tasks, making them difficult to integrate into other TPR-based models.

In this work, we propose a **D**iscrete **D**ictionary-based **D**ecomposition (D3) layer for structured representation learning within the TPR framework. D3 employs the discrete representations techniques to utilize prior knowledge acquired during training for decomposition operations. Inspired by prior discrete key-value architectures [14, 38], D3 consists of multiple dictionaries, each comprising discrete, learnable key-value pairs. Unlike prior work, each dictionary of D3 is linked explicitly to individual TPR components, such as *role*, *filler*, and *unbinding operator*. This design allows each dictionary to capture and store the discrete features of its corresponding TPR components during training. D3 acts as a drop-in layer that maps input data into pre-learned discrete features for the decomposition of TPR components through a three-step process, as illustrated in Fig. 1. First, it generates multiple queries from the input data, with each query utilized for different TPR components. Next, it identifies the nearest codebook keys within each dictionary based on these queries. Finally, D3 generates structured TPR representations by aggregating the codebook values corresponding to these keys. Moreover, D3 can be seamlessly integrated into any TPR-based model by replacing the TPR component generation layer without requiring further modifications.

**Our main contributions** are as follows.

- We propose a novel `D3` layer to tackle the decomposition problem inherent in the TPR-based approaches. `D3` leverages discrete, learnable dictionaries to enhance the decomposition capabilities of TPR-based models. By mapping input data to pre-learned discrete features stored within the dictionaries, `D3` effectively generates structured TPR representations.

- We conduct extensive experiments across various systematic generalization tasks, including synthetic associative recall and text/visual question-answering tasks. Our experimental results show that `D3` significantly enhances the generalization performance of TPR-based models, demonstrating its effectiveness on systematic generalization tasks.

- Our analyses show that `D3` generates well-bound structured representations that are satisfactory for the requirements of the TPR framework, utilizing the discrete, learnable dictionaries.

## 2   Related Work

**Decomposition Problem.**   Compositional generalization in neural networks, which allows for generalizing beyond training data, has been extensively studied [2, 13, 12, 16, 8, 6, 41]. One important capability for achieving this is a *segregation*, as discussed in Greff et al. [6], which enables the formation of meaningful representations from structured and unstructured data [3, 18]. TPR-based neural networks also rely on this capability to generate structured representations for TPR components such as *roles*, *fillers*, and *unbinding operators*. In the TPR framework, these structured representations must satisfy specific conditions to ensure accurate encoding and decoding. First, *roles* need to be linearly independent to avoid *filler* overlap. Second, the *unbinding operator* must correlate with the corresponding *roles* to accurately retrieve associated *fillers*. Recent work [23] has shown that existing TPR-based models often fail to generate structured representations that meet these conditions, undermining their symbolic operations. To address this, an attention-based decomposition module [23] has been introduced, but it still shows limited performance on synthetic tasks involving the decomposition of unseen combinatorial data. In this work, we address the decomposition problem within the TPR framework using a discrete dictionary-based method, advancing the research further.

**Discrete Representation Learning.**   Discrete neural representation learning has introduced a codebook of discrete, learnable representations into neural networks [39]. During training, each discrete representation captures underlying latent features by mapping continuous input data to the nearest features within the codebook, which are then used for downstream tasks. Recent work on object-centric learning has utilized discrete representations to extract specific types of features from unstructured data, leveraging latent features learned during training [11, 43]. Some researchers have proposed a separate key-value codebook for learning discrete representations, demonstrating its effectiveness in systematic generalization [17] and robustness against distributional shifts [38]. Inspired by these findings, we develop a separate key-value-based discrete dictionary method to enhance the decomposition capabilities of TPR-based models. Other researchers have introduced a discrete symbolic embedding layer to improve the interpretability of TPR-based models, showing the feasibility of discrete representations in the TPR framework [21, 9]. However, their methods focus on encoding processes and specific tasks such as question-answering [21] and abstractive summarization [9]. In contrast, our work addresses the decomposition problem in TPR-based approaches, and our `D3` method is a drop-in solution that can be easily adapted to any TPR-based model.

**Memory Network.**   Research on memory networks has focused on enhancing neural network capacity by integrating external memory [36, 4, 5, 24, 27, 41]. Memory-augmented neural networks store variable lengths of sequential data in this external memory and retrieve necessary information using various addressing methods [36, 5]. These writing and reading mechanisms share many similarities with our `D3` approach. However, while memory networks store input features sequentially in their memory states as a continuous stream, `D3` updates symbolic feature information through gradient descent into codebook parameters within dictionaries. This distinctive characteristic allows `D3` to leverage the learned discrete features to decompose unseen data after training. In another work, Lample et al. [14] introduces a learnable key-value memory layer to improve the efficiency of the Transformer by replacing the feed-forward layer. Unlike their memory layer, `D3` employs key-value pairs in dictionaries explicitly linked to individual TPR components, making it well-suited for the TPR framework.

# 3 Method

In this section, we explain how the D3 module generates structured representations of the TPR components using discrete, learnable dictionaries. We then introduce configurations of D3 and how it can be applied to our baseline models.

## 3.1 Discrete Dictionary-based Decomposition module

D3 is a discrete dictionary-based drop-in layer designed to enhance the decomposition capabilities of TPR-based approaches. At every time step, D3 decomposes input data into TPR components, such as *roles*, *fillers*, and *unbinding operators*, by mapping input data to pre-learned symbolic features within dictionaries. These dictionaries consist of discrete, learnable codebook key-value pairs, denoted as $\{\mathcal{D}^j\}_{j=1}^{N_{\text{component}}}$ as shown in Eq. 1. Each dictionary $\mathcal{D}^j$ is explicitly linked to a $j$-th TPR component, allowing it to learn the symbolic features required for generating the specific TPR component. This design also enables the generation of structured representations for different TPR components individually and in parallel.

$$\mathcal{D}^j := \{(\mathsf{k}_i^j, \mathsf{v}_i^j) \mid \mathsf{k}_i^j \in \mathbb{R}^{D_{\text{query}}}, \mathsf{v}_i^j \in \mathbb{R}^{D_{\text{code}}}\}_{i=1}^{N_{\text{code}}} \quad \text{where} \quad j = 1, ..., N_{\text{component}} \tag{1}$$

where $\mathcal{D}^j$ denotes the discrete, learnable dictionary for the $j$-th TPR component, $\mathsf{k}$ denotes a learnable codebook key, and $\mathsf{v}$ denotes a learnable codebook value. In the next paragraph, we describe how D3 generates TPR components using these dictionaries in three steps.

**Step 1: Query Generation.** At each time step $t$, D3 takes input data, denoted as $\texttt{input}_t \in \mathbb{R}^{D_{\text{input}}}$, and generates the query, denoted as $\texttt{queries}_t \in \mathbb{R}^{N_{\text{component}} \times D_{\text{input}}}$, for each $j$-th TPR component using a query network, $f_{\text{query}}^j : \texttt{input}_t \mapsto \texttt{query}_t^j \in \mathbb{R}^{D_{\text{query}}}$. The query network can be any neural network; in this study, we use a feed-forward network with a single layer. Additionally, we apply a layer normalization [1] and a dropout of $p_{\text{dropout}}$ [35] to $\texttt{query}_t^j$.

**Step 2: Sparse Key Access.** D3 searches for the nearest keys from each dictionary, $\mathcal{D}^j$, based on the generated $\texttt{query}_t^j$. We measure the similarity using the inner product between $\texttt{query}_t^j$ and $\{\mathsf{k}_i^j\}_{i=1}^{N_{\text{code}}}$. Then, D3 selects top-$k$ codebook keys in order of largest similarity, as follows.

$$\mathcal{I}^j = \mathcal{T}_k(\texttt{query}_t^{j\top}\hat{\mathsf{k}}_i^j) \quad \text{where} \quad \hat{\mathsf{k}}_i^j = \mathsf{k}_i^j/||\mathsf{k}_i^j||_2 \tag{2}$$

where $\mathcal{T}_k$ denotes the top-$k$ operator that finds the indices of $k$ largest values, and $\mathcal{I}^j$ denotes the indices of the $k$ most similar keys within $\mathcal{D}^j$. We found that applying $L2$ normalization to keys before the inner product mitigates the codebook collapse problem.

**Step 3: Aggregation of Code Values.** D3 computes the normalized score for selected codebook keys, denoted as $w_t^j$, and aggregates codebook values corresponding to selected codebook keys with $w_t^j$, as follows.

$$\texttt{code}_t^j = \Sigma_{i \in \mathcal{I}} w_{t,i}^j \mathsf{v}_i^j \quad \text{where} \quad w_t^j = \text{Softmax}(\texttt{query}_t^{j\top}\hat{\mathsf{k}}_i^j))_{i \in \mathcal{I}^j} \tag{3}$$

Then, D3 maps $\texttt{query}_t^j$ to a dimension of $D_{\text{code}}$ and adds this projected vector to $\texttt{code}_t^j$. The summed vectors are mapped to a dimension of $D_{\text{component}}$ to generate structured representations of TPR components, as follows.

$$\texttt{component}_t^j = \texttt{code}_t^j + \text{layer}_{\text{residual}}(\texttt{query}_t^j) \in \mathbb{R}^{D_{\text{code}}} \tag{4}$$

$$\overline{\texttt{component}}_t^j = \text{layer}_{\text{final}}(\texttt{component}_t^j) \in \mathbb{R}^{D_{\text{component}}} \tag{5}$$

where $\text{layer}_{\text{residual}}$ and $\text{layer}_{\text{final}}$ denote a feed-forward network with a single layer. Those $\overline{\texttt{components}}_t$ are then utilized for TPR operations to solve the downstream tasks.

## 3.2 Module Configurations

In this section, we describe the configurations of D3 when applied to TPR-based models.

**Shared Dictionary between Role and Unbinding Operator.**    As discussed in Section 2, *roles* and *unbinding operators* should have correlated features for accurate TPR operations. Considering this characteristic of the TPR framework, we share the dictionaries of *roles* and *unbinding operators*. This shared dictionary also reduces the number of learnable parameters.

**D3 Applied to Filler.**    While the TPR framework requires specific conditions for *roles* and *unbinding operators* for accurate TPR operations, there are no such requirements for *fillers*. Therefore, we explore two configurations in this study: applying D3 to generate *fillers* (*w/ F*) and not applying D3 to generate fillers (*w/o F*). In the *w/o F* configuration, we follow the baseline models to generate the *filler* representations.

### 3.3   Integration of D3 into Existing TPR-based Models

In this section, we introduce our baseline models and explain how D3 is applied to them, considering the configurations of D3. We use three TPR-based models as our baselines: FWM [30], TPR-RNN [28], and Linear Transformer [10]. Notably, integrating D3 into these baseline models requires only substituting their TPR component generation layer with D3 without further modifications.

**Fast Weight Memory.**    Fast Weight Memory (FWM) [30] is a TPR-based memory network designed for understanding long sequential contexts. It proposes a single word-level TPR operation related to the perceptron learning rule [25]. It has shown significant associative reasoning capability in reinforcement learning and natural language processing tasks. FWM requires two types of *roles* ($role_1$ and $role_2$) and one *filler* for encoding, as well as two types of *unbinding operators* ($unbind_1$ and $unbind_2$) for decoding. When D3 is integrated into FWM, it employs three dictionaries for the shared dictionary configuration: one for the $role_1$ and $unbind_1$, another for the $role_2$ and $unbind_2$, and the other for *filler*, as shown in Fig. 1.

**TPR-RNN.**    TPR-RNN [28] is a sentence-level TPR-based memory network designed for basic text question-answering tasks [42]. It incorporates various encoding operations such as writing, moving, and backlink to process sequential data at the sentence level. These operations necessitate different encoding components with varying dimensions, making direct connections to the decoding components challenging. As a result, we do not apply the shared dictionary configuration to TPR-RNN; instead, we use a shared query network without layer normalization. Furthermore, due to the differing dimensions of the TPR components in TPR-RNN, we employ distinct $layer_{final}$ layers for each TPR component.

**Linear Transformer.**    Linear Transformer [10] linearizes the attention mechanism to improve the computational efficiency of the Transformer [40]. Recently, Schlag et al. [31] demonstrated the equivalence between TPR and the linear attention mechanism, indicating that the key, value, and query in linear attention correspond to the *role*, *filler*, and *unbinding operator*, respectively. Building on this work, we apply D3 to generate the query, key, and value in the Linear Transformer. Unlike TPR-RNN and FWM, the Linear Transformer utilizes multi-head operations. Therefore, we use distinct dictionaries for each head, with the key and query of each head sharing the same dictionary.

## 4   Experiment

In this section, we evaluate the effectiveness of D3 across various tasks, including a synthetic task, text/visual question-answering tasks, and a language modeling task. To assess the decomposition capabilities, we follow the experimental settings of the AID [23], a prior work addressing the decomposition problem in the TPR framework, and closely compare our D3 model to baseline models and AID.

### 4.1   Task

**Systematic Associative Recall (SAR) task.**    This task evaluates systematic generalization in memorizing and recalling combinatorial data [23]. It consists of a discovery phase and an inference phase. During the discovery phase, the model receives the combinatorial sequential items, each combining two symbols, $x \in X$ and $y \in Y$ where $X = X_1 \cup X_2 \cup X_3$ and $Y = Y_1 \cup Y_2$. The

model is then required to predict an associated $y$ when a specific $x$ is presented. The SAR task uses different combination settings between training and evaluation to target systematic generalization specifically. During training, the model learns the following combination settings: (1) $X_1$ and $Y_1$, (2) $X_2$ and $Y_2$, and (3) $X_3$ and $Y$. At the evaluation, on the other hand, the model should generalize unseen combination settings, specifically $X_1$ and $Y_2$. Additionally, the task includes a hyper-parameter $p = \frac{|X_3|}{|X_2|+|X_3|}$ where $|X_i|$ denotes the cardinality of set $X_i$. By adjusting $p$, this task tests the systematic generalization of models under varying levels of exposure to different symbol combinations during training. In our study, we focus solely on the most challenging setting of the SAR task ($p = 0.0$), where the subset $X_3$ is excluded. In the SAR task, the TPR framework regards $x$ as the *role* and the *unbinding operator*, and $y$ as the *filler*. Therefore, TPR-based models should systematically decompose the combinatorial data into structured representations by mapping $x$ to the *role* and $y$ to the *filler* during the discovery phase, and mapping $x$ to the *unbinding operator* during the inference phase to solve this task.

**Systematic bAbI (sys-bAbI) task.** This task is a variant of the bAbI task [42] designed to evaluate systematic generalization in text understanding and reasoning [23]. It consists of 20 distinct sub-tasks, each comprising stories, relevant queries, and corresponding answers. The sys-bAbI task requires the models to remember the stories and predict corresponding answers to the queries. Unlike the original bAbI task, the sys-bAbI task evaluates the models with two aspects: (a) in-distribution (*w/o sys diff*) and (b) with the systematic difference (*w/ sys diff*) where each sub-task includes unseen words during training. Therefore, the models should learn task-independent text understanding to solve the sys-bAbI task.

**Sort-of-CLEVR task.** This task [26] evaluates compositional generalization in visual relational reasoning. It consists of scene images, queries, and corresponding answers. This task requires the models to understand the properties of individual objects (*Unary*) or the relationships between multiple objects (*Binary* or *Ternary*) within visual scene images, and predict the correct answers to the queries [20]. Therefore, the model should capture relationships within each object and between objects to solve this task.

**WikiText-103 task.** This task [19] is a language modeling dataset consisting of lengthy corpora from Wikipedia. Although the WikiText-103 task does not directly measure the systematic generalization of the models, it is used to evaluate the effectiveness and applicability of D3 on a large-scale task beyond relatively simple tasks.

## 4.2 Experimental Results

In this section, we present the experimental results of the SAR task, sys-bAbI task, sort-of-CLEVR task, and WikiText-103 task. In our experiments, we set $D_{\text{query}}$ as $D_{\text{code}}/2$.

### 4.2.1 TPR-based Memory Networks

First, we evaluate FWM with D3 on the SAR task, which requires understanding the composition of two types of symbols, $x$ and $y$. TPR-based models are expected to solve this task perfectly by mapping each symbol to a specific TPR component during decomposition. However, as shown in Fig. 2, FWM and AID fail to generalize unseen combinations of known symbols. In contrast, our D3 module significantly outperforms other baseline models, achieving nearly 100% accuracy. This result demonstrates that D3 effectively decomposes unseen combinatorial data into TPR components using discrete dictionaries.

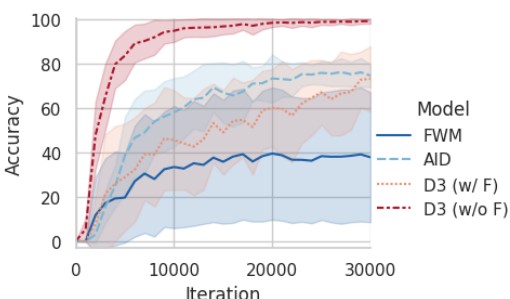

Figure 2: Test accuracy curve [%] on the SAR task for 10 seeds, with shadowed area indicating SD.

Next, we test TPR-RNN and FWM with D3 on the sys-bAbI task. This task involves compositional information in each story sentence, such as the relation between objects and their locations. It makes

Table 1: The mean word error rate [%] on the sys-bAbI task for 10 seeds, with $\pm$ indicating SD.

| Model | w/o sys diff ($\downarrow$) | w/ sys diff ($\downarrow$) | Gap ($\downarrow$) | # params ($\downarrow$) |
|---|---|---|---|---|
| TPR-RNN | $0.79 \pm 0.16$ | $8.74 \pm 3.74$ | 7.95 | **0.14** $M$ |
| + AID | $\underline{0.69} \pm 0.08$ | $\underline{5.61} \pm 1.78$ | $\underline{4.92}$ | 0.32 $M$ |
| + D3 | $\textbf{0.65} \pm 0.25$ | $\textbf{3.50} \pm 2.07$ | **2.85** | $\underline{0.17}$ $M$ |
| FWM | $0.79 \pm 0.14$ | $2.85 \pm 1.61$ | 2.06 | **0.73** $M$ |
| + AID | $\textbf{0.45} \pm 0.16$ | $\textbf{1.21} \pm 0.66$ | **0.76** | 1.23 $M$ |
| + D3 (w/o F) | $0.79 \pm 0.30$ | $2.58 \pm 1.12$ | 1.79 | $\underline{0.75}$ $M$ |
| + D3 (w/ F) | $\underline{0.75} \pm 0.17$ | $\underline{1.96} \pm 0.88$ | $\underline{1.21}$ | $0.75$ $M$ |

Table 2: The mean accuracy [%] on the sort-of-CLEVR task for 10 seeds, with $\pm$ indicating SD.

| Model | $D_{\text{code}}$ | Unary ($\uparrow$) | Binary ($\uparrow$) | Ternary ($\uparrow$) | # params ($\downarrow$) |
|---|---|---|---|---|---|
| Linear Transformer | - | $69.3 \pm 14.8$ | $75.5 \pm 1.3$ | $56.4 \pm 4.3$ | **0.68** $M$ |
| + AID | - | $\underline{98.9} \pm 0.2$ | $78.6 \pm 0.3$ | $\underline{63.7} \pm 1.2$ | 0.83 $M$ |
| + D3 (w/o F) | 128 | $73.9 \pm 16.5$ | $77.2 \pm 2.2$ | $57.3 \pm 4.6$ | $\underline{0.75}$ $M$ |
| | 256 | $73.7 \pm 16.5$ | $77.8 \pm 2.5$ | $57.9 \pm 5.8$ | 0.96 $M$ |
| + D3 (w/ F) | 128 | $\underline{98.9} \pm 0.2$ | $\underline{79.5} \pm 0.8$ | $63.1 \pm 1.9$ | 0.80 $M$ |
| | 256 | $\textbf{99.0} \pm 0.3$ | $\textbf{82.1} \pm 2.4$ | $\textbf{68.8} \pm 1.2$ | 1.13 $M$ |

Table 3: Perplexity on the WikiText-103 task.

| Model | $D_{\text{code}}$ | Valid ($\downarrow$) | Test ($\downarrow$) | # params ($\downarrow$) |
|---|---|---|---|---|
| Linear Transformer | - | 36.473 | 37.533 | **44.02** $M$ |
| + AID | - | 36.159 | 37.151 | 44.16 $M$ |
| + D3 (w/o F) | 32 | $\underline{36.061}$ | 37.220 | $\underline{44.12}$ $M$ |
| | 64 | **35.975** | **37.009** | 44.36 $M$ |
| + D3 (w/ F) | 32 | 36.630 | 37.620 | 44.22 $M$ |
| | 64 | 36.220 | $\underline{37.128}$ | 44.62 $M$ |

a sentence-level model more suitable for capturing the structural information of data than a word-level model. However, as shown in Table 1, TPR-RNN shows a larger performance gap between the *w/o sys diff* and *w/ sys diff* cases than FWM. Notably, D3 enhances the systematic generalization of both TPR-RNN and FWM with fewer additional parameters, significantly reducing the performance gap for TPR-RNN. These results highlight the efficacy of D3 in text understanding tasks.

### 4.2.2 Linear Transformer

We also evaluate the Linear Transformer with D3 on the sort-of-CLEVR task and WikiText-103 task. Following the AID [23], we use a 4-layered Linear Transformer with shared parameters for the sort-of-CLEVR task and apply D3 to a 16-layered Linear Transformer at intervals of 4 out of the 16 layers for the WikiText-103 task. As shown in Tables 2 and 3, D3 improves the performance of the Linear Transformer, with these improvements increasing as the capacity of the dictionaries grows. These results demonstrate the effectiveness of D3 on visual relational reasoning and language modeling tasks, as well as its applicability to the Linear Transformer. In addition, D3 shows comparable performance to the attention-based decomposition method, even with fewer parameters.

### 4.3 Analysis

In this section, we conduct a qualitative analysis of the structured TPR representations generated by D3 and an ablation study of D3. For these analyses, we experiment with D3 (*w/o F*) on the SAR task.

### 4.3.1 Qualitative Analysis

TPR framework requires its structured representations to satisfy the following conditions for accurate TPR operations: (*i*) linearly independence between distinct *roles*, and (*ii*) high correlation between

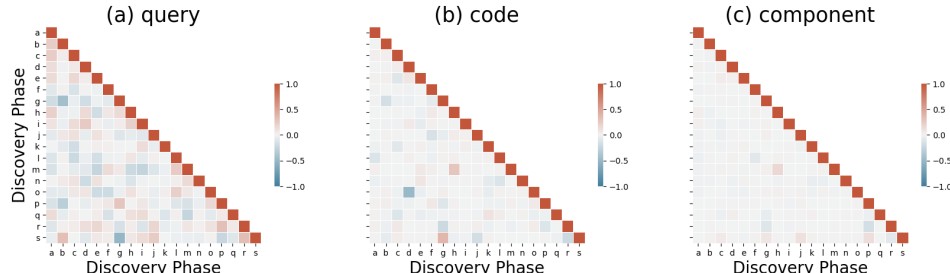

Figure 3: The heatmap displays the cosine similarity between the generated representations during the discovery phase for the SAR task. We explore the similarity across different types of representations: (a) `queries` of *roles*, (b) `codes` of *roles*, and (c) the *roles* themselves.

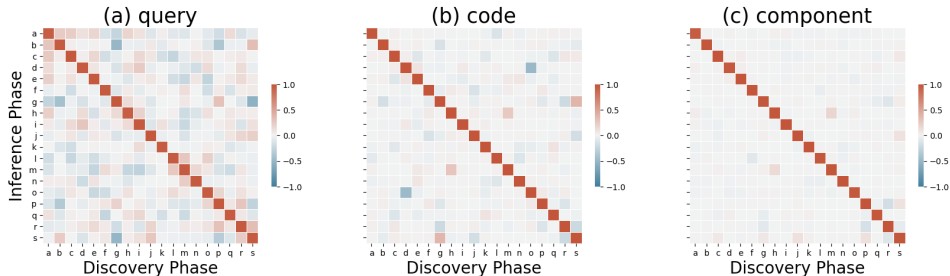

Figure 4: The heatmap displays the cosine similarity between the generated representations during the discovery phase (represented on the **x-axis**) and the inference phase (represented on the **y-axis**) for the SAR task. We explore the similarity across different types of representations: (a) `queries` of *roles* and *unbinding operators*, (b) `codes` of *roles* and *unbinding operators*, and (c) the *roles* and *unbinding operators* themselves.

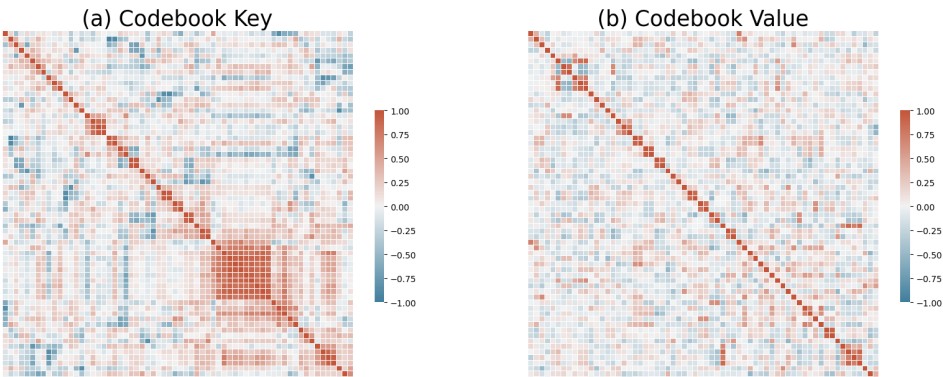

Figure 5: The heatmap visualizes the cosine similarity of the learned codebook features for the SAR task. There are two parts to each heatmap: (a) the similarity among codebook keys, denoted as $\{k_i\}_{i=1}^{N\text{code}}$, and (b) the similarity among codebook values, denoted as $\{v_i\}_{i=1}^{N\text{code}}$. For better visualization, the heatmap values are reordered to reflect the cluster of similar codebook keys.

*role* and *unbinding operator* for the same symbol $x$. We analyze the orthogonality of generated representations to investigate whether they satisfy these TPR conditions. Specifically, we consider the case of varying $x$ while keeping $y$ fixed for simplicity.

Fig. 3(c) shows the cosine similarity between the *roles* during the discovery phase, and Fig. 4(c) shows the cosine similarity between the *roles* during the discovery phase and the *unbinding operator* during the inference phase. Both results demonstrate that the generated representations by D3 satisfy the TPR conditions, resulting in an accuracy of nearly 100%. We also conduct the same analysis for intermediate features, particularly `query` and `code`. Figs. 3 and 4 show that each intermediate representation complements the others to satisfy the TPR condition, indicating the effectiveness of D3.

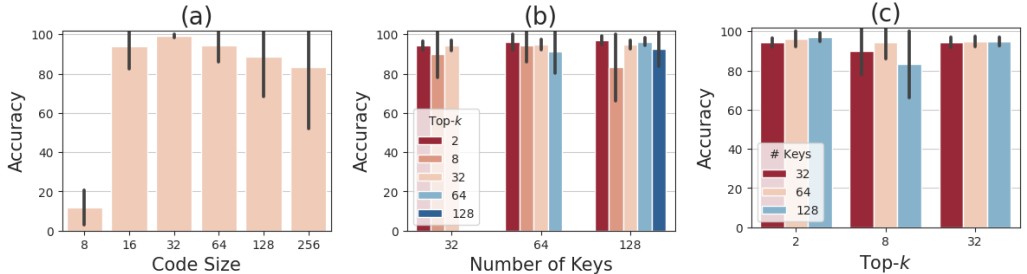

Figure 6: The mean accuracy on the SAR task for 10 seeds in the ablation study, with error bar indicating SD. The default setting uses $D_{\text{code}}$ of 64, $N_{\text{code}}$ of 64, and top-$k$ of 8. Each figure shows the experimental results for the following settings: (a) Varying $D_{\text{code}}$. (b) Varying $N_{\text{code}}$ with top-$k$ constant. (c) Varying top-$k$ with $N_{\text{code}}$ constant.

Furthermore, we analyze the similarity patterns of codebook keys and codebook values. Fig. 5 shows that the codebook features learn orthogonal patterns despite being learned without constraints. This result implies that the learnable parameters of dictionaries implicitly capture TPR conditions to ensure accurate TPR operations.

### 4.3.2 Ablation Study

We investigate the effect of hyper-parameters of D3, specifically $N_{\text{code}}$, $D_{\text{code}}$, and top-$k$, on performance on the SAR task. Fig. 6(a) shows the effect of $D_{\text{code}}$. We observe that the value of $D_{\text{code}}$ significantly affects the performance of D3. Notably, D3 fails to solve the SAR task when $D_{\text{code}}$ is set to 8, indicating a need for adequate capacity of $D_{\text{code}}$. Fig. 6(b) shows the effect of varying top-$k$ while holding $N_{\text{code}}$ constant, indicating that D3 achieves optimal performance when top-$k$ is set to 2. This result demonstrates the efficacy of the sparse mechanism employed by D3. Fig. 6(c) examines the effect of varying $N_{\text{code}}$ while holding top-$k$ constant, showing that D3 generally performs better with larger values of $N_{\text{code}}$.

## 5 Discussion and Limitations

**Motivation.** From the perspective of systematic generalization, the decomposition operations in the TPR framework can be viewed as mapping unseen data to TPR components observed during training. Motivated by this, we design a decomposition module based on discrete representations, which maps input data to discrete, learned features facilitating systematic generalization in the decomposition operations of TPR. This design choice differentiates our contribution from AID's competitive attention-based decomposition module. Additionally, each dictionary in D3 is explicitly linked to a specific TPR component, ensuring that each dictionary is responsible solely for generating its corresponding component. The generated components are then utilized in predefined TPR operations of the TPR-based models. This design ensures that each dictionary is trained to specialize in a specific TPR component.

**Interpretability.** The TPR framework decomposes data at the representation level into distinct symbols, such as *role-filler* pairs for encoding and *unbinding operators* for decoding. This characteristic enhances the interpretability of models because the relationships between *roles* and *unbinding operators* explain which parts of the input the model focuses on to predict the output. However, this interpretability is reliable only when the generated structured representations satisfy the TPR conditions. In this context, D3 enhances the interpretability of models by providing structured representations that more effectively satisfy the TPR conditions than baseline models like FWM and AID. Figs. 9 and 10 demonstrate that the representations generated by D3 better conform to the TPR conditions than those from other baseline models, supporting our claim that D3 contributes to increased interpretability.

D3 **Applied to Filler (w/o F and w/ F).** In the TPR framework, *roles* and *unbinding operators* must meet specific conditions, such as linear independence among *roles* and high correlation between *roles* and *unbinding operators*, to ensure accurate TPR operations. However, there are no such

requirements for *fillers*, which are features related to downstream tasks. This characteristic affects the performance of D3 depending on whether it is applied to generate the *fillers* (*w/ F*) or not (*w/o F*). In our experiments, the *w/ F* configuration performs well on the sys-bAbI and sort-of-CLEVR tasks with relatively few labels (~200). In contrast, the *w/o F* configuration excels on the SAR and WikiText-103 tasks, which have a larger number of labels (500~). These findings suggest that the *w/o F* configuration may be more effective for large-scale practical tasks. Nevertheless, beyond these experimental results, we do not fully understand the conditions under which each configuration performs better. Consequently, one limitation of D3 is the additional burden of determining the suitable configuration for various tasks when applying it to other domains.

**Sparse Key Selection.** D3 integrates seamlessly with existing TPR-based models, significantly enhancing their generalization performance across various tasks. However, this integration introduces additional computational overhead to the baseline models. Specifically, the sparse key selection mechanism of D3 has a computational complexity of $\mathcal{O}(N_{\text{code}} \times (D_{\text{query}} + \log k))$ for each TPR component. Therefore, this complexity can become a drawback as the capacity of the dictionaries increases. One potential solution to address this capacity issue is to incorporate product keys into the sparse key selection mechanism of D3, a technique studied in prior discrete key-value architectures [14]. We leave this enhancement for future work.

**Scalability.** The scalability of D3 is inherently linked to TPR operations of baseline models since the number of dictionaries in the D3 layer aligns with the number of TPR components required for their operations. As TPR operations require increasing components to handle large datasets, our method also requires a proportional increase in dictionaries, resulting in significant computational and memory overhead. As explored in prior work, one potential solution to mitigate this issue is distributing shared dictionaries across multiple heads or layers [14]. However, this approach requires further investigation and experimentation, which we plan to research in future work.

# 6  Conclusion

In this paper, we tackle the decomposition problem inherent in the TPR framework, which poses a significant challenge for TPR-based models. To address this, we introduce a discrete dictionary-based layer, D3, designed to enhance the decomposition capabilities of TPR-based models. D3 employs the discrete dictionaries to map input data to pre-learned symbolic features within each dictionary, thereby generating structured TPR representations. Our comprehensive experiments demonstrate that D3 significantly enhances the systematic generalization of the TPR-based models with fewer additional parameters. Furthermore, our qualitative analysis verifies that D3 effectively generates structured representations that are satisfactory for the requirements of the TPR framework.

## Acknowledgements

This work was supported by the National Research Foundation of Korea (NRF) grant funded by the Korea government (MSIT) (No. 2021R1A2C3011169 & No. 2022R1A5A7026673 & No. RS-2022-00166735 & No. RS-2023-00218987).

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

# Appendix

## A  Experiment Details

This section provides a detailed description of our experiments on the SAR task, sys-bAbI task, sort-of-CLEVR task, and WikiText-103 task. We followed the experimental settings outlined by AID [23] to assess the decomposition capabilities of D3. To ensure stability and reproducibility, we ran all experiments, except for the WikiText-103 task, using 10 different random seeds[3]. For the WikiText-103 task, we experimented with a single seed of 1111. Each experiment was conducted on a single 48GB NVIDIA RTX A6000 GPU and an AMD EPYC 7513 32-Core Processor.

### A.1  Systematic Associative Recall task

The SAR task [23] evaluates systematic generalization in memorizing and recalling combinatorial data. It consists of a discovery phase and an inference phase. During the discovery phase, the model receives the combinatorial sequential items, each combining two symbols, $x \in X$ and $y \in Y$ where $X = X_1 \cup X_2 \cup X_3$ and $Y = Y_1 \cup Y_2$. The model is then required to predict an associated $y$ when a specific $x$ is presented. The SAR task uses different combination settings between training and evaluation to target systematic generalization specifically. During the training, the model learns the following combination settings: (1) $X_1$ and $Y_1$, (2) $X_2$ and $Y_2$, and (3) $X_3$ and $Y$. At evaluation, however, the model should generalize unseen combination settings, specifically $X_1$ and $Y_2$. In our study, unlike the AID paper [23], we only consider the most challenging setting of the SAR task by excluding the subset $X_3$.

Each combinatorial item is constructed as follows. First, symbols $x$ and $y$ are sampled from their respective sets $X$ and $Y$, where $|X_1| = |X_2| = |Y_1| = |Y_2| = 250$. The sampled symbols are mapped into a 50-dimensional space using a word embedding method. These embedding vectors are then concatenated to construct the combinatorial item. For training, 100 randomly generated combinatorial items are sequentially provided to the model during the discovery phase. During the inference phase, the model receives only the $x$ symbols sequentially, with the embedding vector of $y$ set to zero. This task also provides binary flags to indicate the start of each phase. At evaluation, all possible combinations that can be formed in $X_1$ and $Y_2$ are tested.

To build the experimental environment for the SAR task, we utilize the open-source implementation[4] from the AID [23]. We train the model using the Adam optimizer with a batch size of 64 and a learning rate of $1e^{-3}$, $\beta_1$ of 0.9, and $\beta_2$ of 0.98 for training iterations of $30K$. Each experiment took approximately 3 hours per each seed.

### A.2  Systematic bAbI task

The sys-bAbI task [23] is a variant of the bAbI task [42] designed to evaluate systematic generalization in text understanding and reasoning. It consists of 20 distinct sub-tasks, each comprising stories, relevant queries, and corresponding answers. The sys-bAbI task requires the models to remember the stories and predict corresponding answers to the queries. Unlike the original bAbI task, the sys-bAbI task evaluates the models with two aspects: (a) in-distribution (*w/o sys diff*) and (b) with the systematic difference (*w/ sys diff*) where each sub-task includes unseen words during training. Therefore, the models should learn task-independent text understanding to solve the sys-bAbI task.

The bAbI dataset includes various versions, such as `en-10k` and `en-valid-10k`. The sys-bAbI task uses the `en-valid-10k` version, which is already divided into training, validation, and test datasets. To create the experimental environment for the sys-bAbI task, we use the open-source implementation[5] provided by the AID.

---

[3]We used the following seed values: {0, 1111, 2222, 3333, 4444, 5555, 6666, 7777, 8888, 9999}

[4]https://github.com/taewonpark/AID/tree/main/SARtask

[5]https://github.com/taewonpark/AID/tree/main/bAbItask

We use the open-source implementation of the baseline models, TPR-RNN[6] [28] and FWM[7] [30]. Following the experimental settings of baseline models, we use different configurations for each model. We train the TPR-RNN with D3 using an embedding size of 179 and the Adam optimizer with a batch size of 128 and a learning rate of $1e^{-3}$, $\beta_1$ of 0.9, and $\beta_2$ of 0.99 for 100 training epochs. For FWM with D3, we use an embedding size of 256 and the Adam optimizer with a batch size of 64 and a learning rate of $1e^{-3}$, $\beta_1$ of 0.9, and $\beta_2$ of 0.98 for training iterations of $60K$. Furthermore, following the AID, we use the reconstruction loss for the bAbI task, introduced in Park et al. [22], in our experiments on the sys-bAbI task. Each experiment took approximately 7 hours per seed for the TPR-RNN with D3 and 8 hours per seed for the FWM with D3.

## A.3 Sort-of-CLEVR task

The sort-of-CLEVR task [26] evaluates compositional generalization in visual relational reasoning. It consists of scene images, queries, and corresponding answers. This task requires the models to understand the properties of individual objects (*Unary*) or the relationships between multiple objects (*Binary* or *Ternary*) within visual scene images and predict the correct answers to the queries. Therefore, the model should capture relationships within each object and between objects to solve this task.

Each scene image, with a size of $75 \times 75$ pixels, includes 6 distinct objects in 6 different colors (red, blue, green, orange, yellow, or gray) and 2 different shapes (square or circle). This scene image is encoded by a visual encoder. The encoded visual feature is then concatenated with the embedding vector of the query. These concatenated features are provided to the model. Following the experimental settings of the AID [23], we use a single CNN layer with a kernel size of 15 and a stride of 15 for the visual encoder, and an embedding size of 128 for the word embedding method. Also, we use a 4-layered Transformer, where each layer shares its parameters with others, as our baseline model.

To build the experimental environment for the sort of CLEVR task, we utilize the open-source implementation[8] from Mittal et al. [20]. We train the model using the Adam optimizer with a batch size of 64 and a learning rate of $1e^{-4}$ for 100 training epochs. Each experiment took approximately 2.5 hours per each seed.

## A.4 WikiText-103 task

The WikiText-103 task [19] is a language modeling dataset consisting of lengthy corpora from Wikipedia. Although the WikiText-103 task does not directly measure the systematic generalization of the models, it is used to evaluate the effectiveness and applicability of D3 on a large-scale task beyond relatively simple tasks.

The WikiText-103 task comprises 28,475 articles for training, 60 for validation, and 60 for testing. Following the experimental settings of Schlag et al. [31], we partition the articles into segments of $L$ words. During training, the gradient is back-propagated only within spans of $L$ words. The performance of the model is evaluated using the measure of perplexity. During evaluation, the model processes an input sequence of $L$ words by sliding a segment over the article with a stride size of 1. Perplexity is then computed based on the last position of each segment, except for the first segment, where every position is taken into account.

To build the experimental environment for the WikiText-103 task, we utilize the open-source implementation[9] from [31]. Following the AID [23], we apply D3 to a 16-layered Linear Transformer at intervals of 4 out of the 16 layers. We train the model using the Adam optimizer with a batch size of 96, an initial learning rate of $2.5e^{-4}$, and a learning rate warmup step of 2,000 for 120 epochs. Each experiment took approximately ~3 days.

---

[6]https://github.com/APodolskiy/TPR-RNN-Torch
[7]https://github.com/ischlag/Fast-Weight-Memory-public
[8]https://github.com/sarthmit/Compositional-Attention/tree/main/Sort-of-CLEVR
[9]https://github.com/IDSIA/lmtool-fwp

# B Hyper-parameter Settings

Table 4: Hyper-parameter settings of the D3.

|  | **SAR task** | *sys-bAbI* **task** | **Sort-of-CLEVR task** | **WikiText-103 task** |
|---|---|---|---|---|
| $D_{\text{code}}$ | 8, 16, 32, 64, 128 | 32, 64, 128, 256 | 128, 256 | 32, 64 |
| $N_{\text{code}}$ | 64 | | | |
| $D_{\text{query}}$ | $D_{\text{code}}/2$ | | | |
| top-$k$ | 8 | | | |
| $p_{\text{dropout}}$ | 0.1 | | | |

Table 5: Hyper-parameters of TPR-RNN.

|  | *sys-bAbI* **task** |
|---|---|
| $D_{\text{entity}}$ ($D_{\text{component}}$) | 90 |
| $D_{\text{relation}}$ ($D_{\text{component}}$) | 20 |
| $N_{\text{component}}^{\text{enc}}$ | 5 |
| $N_{\text{component}}^{\text{dec}}$ | 4 |

Table 6: Hyper-parameters of FWM.

|  | **SAR task** | *sys-bAbI* **task** |
|---|---|---|
| $D_{\text{LSTM}}$ | 256 | 256 |
| $D_{\text{FWM}}$ ($D_{\text{component}}$) | 32 | 32 |
| $N_{\text{reads}}$ | 1 | 3 |
| $N_{\text{component}}^{\text{enc}}$ | 3 | 3 |
| $N_{\text{component}}^{\text{dec}}$ | $1+N_{\text{reads}}$ | $1+N_{\text{reads}}$ |

Table 7: Hyper-parameters of Linear Transformer.

|  | **Sort-of-CLEVR task** | **WikiText-103 task** |
|---|---|---|
| $D_{\text{heads}}$ ($D_{\text{component}}$) | 64 | 16 |
| $N_{\text{heads}}$ | 4 | 8 |
| $N_{\text{component}}^{\text{enc}}$ | 2 * $N_{\text{heads}}$ | 2 * $N_{\text{heads}}$ |
| $N_{\text{component}}^{\text{dec}}$ | $N_{\text{heads}}$ | $N_{\text{heads}}$ |

## C Additional Experiments

Table 8: The mean word error rate [%] on additional experiments of the *sys-bAbI* task for 10 seeds.

| Model | $D_{\text{code}}$ | *w/o sys diff* ($\downarrow$) | *w/ sys diff* ($\downarrow$) | **Gap** ($\downarrow$) | # params ($\downarrow$) |
|---|---|---|---|---|---|
| TPR-RNN | - | $0.79 \pm 0.16$ | $8.74 \pm 3.74$ | 7.95 | $\underline{0.14}\ M$ |
| + AID | - | $0.69 \pm 0.08$ | $5.61 \pm 1.78$ | 4.92 | $0.32\ M$ |
| + D3 | 32 | $1.16 \pm 0.25$ | $\mathbf{3.44} \pm 1.78$ | $\mathbf{2.28}$ | $\mathbf{0.13}\ M$ |
| | 64 | $\mathbf{0.65} \pm 0.25$ | $\underline{3.50} \pm 2.07$ | $\underline{2.85}$ | $0.17\ M$ |
| | 128 | $\underline{0.68} \pm 0.14$ | $3.94 \pm 2.20$ | 3.26 | $0.26\ M$ |
| FWM | - | $0.79 \pm 0.14$ | $2.85 \pm 1.61$ | 2.06 | $\mathbf{0.73}\ M$ |
| + AID | - | $\mathbf{0.45} \pm 0.16$ | $\mathbf{1.21} \pm 0.66$ | $\mathbf{0.76}$ | $1.23\ M$ |
| + D3 (*w/o F*) | 64 | $0.79 \pm 0.30$ | $2.58 \pm 1.12$ | 1.79 | $0.75\ M$ |
| | 128 | $0.93 \pm 0.20$ | $3.82 \pm 1.21$ | 2.89 | $0.82\ M$ |
| | 256 | $1.04 \pm 0.40$ | $3.33 \pm 1.21$ | 2.29 | $0.97\ M$ |
| + D3 (*w/ F*) | 32 | $1.20 \pm 0.31$ | $7.23 \pm 4.33$ | 6.03 | $\underline{0.71}\ M$ |
| | 64 | $\underline{0.75} \pm 0.17$ | $\underline{1.96} \pm 0.88$ | $\underline{1.21}$ | $0.75\ M$ |
| | 128 | $0.89 \pm 0.32$ | $2.48 \pm 0.67$ | 1.59 | $0.84\ M$ |
| | 256 | $\underline{0.75} \pm 0.23$ | $3.09 \pm 1.83$ | 2.34 | $1.02\ M$ |

## D Additional Comparisons

In this section, we expand our comparisons to include a broader range of state-of-the-art methods, as detailed below.

**sys-bAbI task.** We compare D3 to state-of-the-art methods (DAM [22] and STM [15]) on the original bAbI task. Table 9 shows that existing memory networks struggle with the sys-bAbI task, highlighting the efficacy of D3 compared to these state-of-the-art memory networks.

Table 9: The mean word error rate [%] on additional comparison of the sys-bAbI task for 10 seeds.

| Model | *w/o sys diff* ($\downarrow$) | *w/ sys diff* ($\downarrow$) | **Gap** ($\downarrow$) |
|---|---|---|---|
| DAM | $0.48 \pm 0.20$ | $5.25 \pm 1.64$ | 4.77 |
| STM | $0.49 \pm 0.16$ | $4.79 \pm 1.53$ | 3.70 |
| TPR-RNN | $0.79 \pm 0.16$ | $8.74 \pm 3.74$ | 7.95 |
| + AID | $\underline{0.69} \pm 0.08$ | $\underline{5.61} \pm 1.78$ | $\underline{4.92}$ |
| + D3 | $\mathbf{0.65} \pm 0.25$ | $\mathbf{3.50} \pm 2.07$ | $\mathbf{2.85}$ |
| FWM | $0.79 \pm 0.14$ | $2.85 \pm 1.61$ | 2.06 |
| + AID | $\mathbf{0.45} \pm 0.16$ | $\mathbf{1.21} \pm 0.66$ | $\mathbf{0.76}$ |
| + D3 (*w/o F*) | $0.79 \pm 0.30$ | $2.58 \pm 1.12$ | 1.79 |
| + D3 (*w/ F*) | $\underline{0.75} \pm 0.17$ | $\underline{1.96} \pm 0.88$ | $\underline{1.21}$ |

**Sort-of-CLEVR task.** We compare D3 to vanilla Transformer [40] and Compositional Transformer [20], designed to enhance the systematic generalization capabilities of multi-head self-attention methods. Table 10 shows that the Linear Transformer significantly degrades systematic generalization performance compared to the vanilla Transformer and the Compositional Transformer. While D3 improves the performance of the Linear Transformer from a TPR perspective, it still shows limited performance in reasoning the relationships between multiple objects (*Binary* and *Ternary*) compared to the vanilla Transformer and Compositional Transformer.

**WikiText-103 task.** We compared D3 to the Delta Network [31], which introduced a delta updating rule instead of the additive outer product-based updating rule in the Linear Transformer. Table 11

Table 10: The mean accuracy [%] on additional comparison of the sort-of-CLEVR task for 10 seeds.

| Model | $D_{code}$ | Unary (↑) | Binary (↑) | Ternary (↑) |
|---|---|---|---|---|
| Transformer | - | 97.4 ± 3.5 | 84.3 ± 4.3 | 62.7 ± 3.9 |
| Compositional Transformer | - | 98.9 ± 0.2 | **88.4** ± 1.4 | 66.5 ± 1.9 |
| Linear Transformer | - | 69.3 ± 14.8 | 75.5 ± 1.3 | 56.4 ± 4.3 |
| + AID | - | 98.9 ± 0.2 | 78.6 ± 0.3 | 63.7 ± 1.2 |
| + D3 (w/o F) | 128 | 73.9 ± 16.5 | 77.2 ± 2.2 | 57.3 ± 4.6 |
| | 256 | 73.7 ± 16.5 | 77.8 ± 2.5 | 57.9 ± 5.8 |
| + D3 (w/ F) | 128 | 98.9 ± 0.2 | 79.5 ± 0.8 | 63.1 ± 1.9 |
| | 256 | **99.0** ± 0.3 | 82.1 ± 2.4 | **68.8** ± 1.2 |

indicates that although D3 improves the performance of the Linear Transformer in language modeling tasks, the choice of updating rules has a more substantial impact on performance for tasks involving the comprehension of lengthy corpora than the decomposition operation.

Table 11: Perplexity on additional comparison of the WikiText-103 task.

| Model | $D_{code}$ | Valid (↓) | Test (↓) |
|---|---|---|---|
| Delta Network | - | **35.640** | **36.659** |
| Linear Transformer | - | 36.473 | 37.533 |
| + AID | - | 36.159 | 37.151 |
| + D3 (w/o F) | 32 | 36.061 | 37.220 |
| | 64 | 35.975 | 37.009 |
| + D3 (w/ F) | 32 | 36.630 | 37.620 |
| | 64 | 36.220 | 37.128 |

# E  Additional Ablation Study

In this section, we extend our ablation studies to investigate the effects of varying the number of keys in the codebook and the impact of removing either the residual connection or the codebook from the D3 layer.

**The Effect of Varying the Number of Codebook Keys.**  Fig. 7 shows that even with a significantly reduced number of keys, the model with D3 maintains high accuracy on the SAR task. This observation prompts the question of how consistent performance is achieved despite the reduction in codebook size. To explore this further, we examine the impact of removing the codebook or the residual connection within the D3 layer on the SAR and sys-bAbI tasks. Specifically, removing the codebook means that the components are generated solely by the shared feed-forward networks (layer$_{residual}$ and layer$_{final}$) while removing the residual connection implies that the components are derived solely from the codebook values.

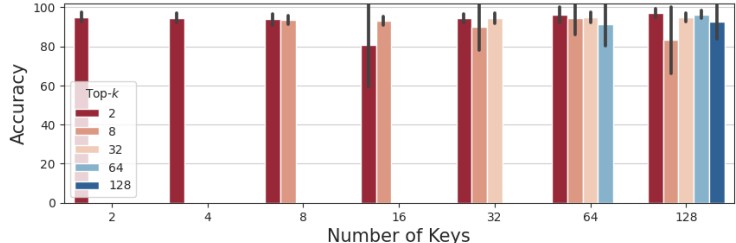

Figure 7: The mean accuracy on the SAR task for 10 seeds in the ablation study for the effect of varying $N_{code}$ from 2 to 128 with top-$k$ constant.

**The Effect of Residual Connection.** Fig. 8 shows that without the residual connection, the generalization performance of D3 dramatically degrades. This result indicates that the residual connection is crucial for effectively training the D3 layer.

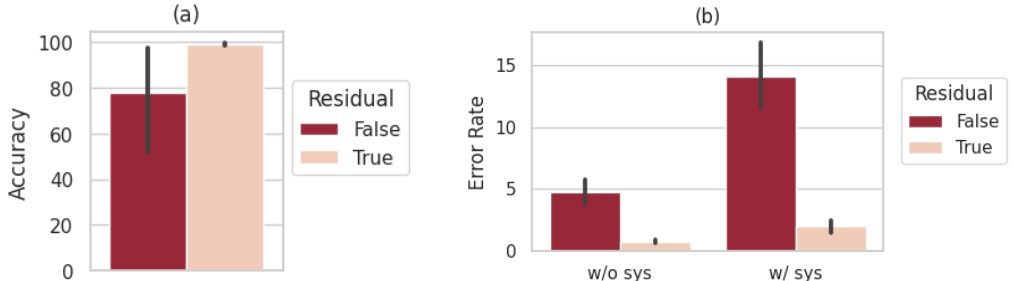

Figure 8: Ablation study for the effect of the residual connection on (a) the SAR task and (b) the sys-bAbI task for 10 seeds.

**The Effect of Codebook.** Table 12 shows that even without the codebook ("*w/o codebook*"), the D3 layer improves the generalization performance of the baseline model on the SAR task. This result indicates that the shared feed-forward networks significantly contribute to performance enhancement, which may explain why the model maintains robust performance even with fewer keys.

However, it is important to note that without the codebook, the D3 layer does not achieve near-perfect accuracy on the SAR task (as shown in Table 12) and fails to significantly enhance the systematic generalization of the baseline model on the sys-bAbI task (as shown in Table 13). These results demonstrate that the codebook plays a crucial role in enhancing the model's overall performance and generalization capabilities, especially in tasks requiring systematic generalization.

Furthermore, we experiment with $N_{code} = 1$ on the SAR task, where the codebook may act as a bias term. The results in Table 12) show that using a single codebook element leads to degraded generalization performance compared to the "*w/o codebook*" configuration, indicating that multiple codebook elements are essential for achieving optimal results.

Table 12: Ablation study for the effect of the codebook on the SAR task for 10 seeds.

| Model | $D_{code}$ | $N_{code}$ | top-$k$ | Accuracy ($\uparrow$) |
|---|---|---|---|---|
| FWM | - | - | - | 44.90 $_{\pm 31.5}$ |
| + D3 |  | 4 | 2 | 87.38 $_{\pm 11.10}$ |
| + D3 | 32 | 64 | 8 | **99.27** $_{\pm 0.88}$ |
| + D3 (*w/o codebook*) |  | - | - | 89.02 $_{\pm 4.56}$ |
| + D3 |  | 1 | 1 | 89.10 $_{\pm 7.99}$ |
| + D3 |  | 4 | 2 | **94.47** $_{\pm 2.35}$ |
| + D3 | 64 | 64 | 8 | 94.29 $_{\pm 8.06}$ |
| + D3 (*w/o codebook*) |  | - | - | 91.65 $_{\pm 3.66}$ |

Table 13: Ablation study for the effect of the codebook on the sys-bAbI task for 10 seeds.

| Model | *w/o sys diff* ($\downarrow$) | *w/ sys diff* ($\downarrow$) | Gap ($\downarrow$) |
|---|---|---|---|
| FWM | 0.79 $_{\pm 0.14}$ | 2.85 $_{\pm 1.61}$ | 2.06 |
| + D3 | **0.75** $_{\pm 0.17}$ | **1.96** $_{\pm 0.88}$ | **1.21** |
| + D3 (*w/o codebook*) | 1.19 $_{\pm 0.41}$ | 3.55 $_{\pm 1.04}$ | 2.36 |

**Discussion.** Our ablation study on the codebook in the SAR task (Table 12) indicates that the shared residual networks within the D3 layer significantly enhance generalization performance. However, the results from the sys-bAbI task (Table 13) suggest that while these networks improve performance, they alone struggle to generalize more structured data.

The ablation studies in Tables 12 and 13 demonstrate that incorporating the codebook mechanism leads to nearly 100% accuracy on the SAR task and significantly improves the systematic generalization of models on the sys-bAbI task. However, as shown in the ablation study on the residual connection (Fig. 8), the codebook alone does not achieve the same level of generalization and exhibits instability within the D3 layer.

In conclusion, our experimental results indicate that the combination of the codebook and the shared residual networks within the D3 layer is crucial for enhancing systematic generalization performance and stability. By integrating these two components, our D3 layer significantly improves the systematic generalization capabilities of TPR-based models.

# F   Additional Qualitative Analysis

## F.1   Comparison to Baselines

We conduct an orthogonal analysis for the baseline models (FWM and AID) similar to the analysis presented in Section 4.3.1. Figs. 9 and 10 indicate that the D3 model generates more structured and orthogonal representations than the baseline models, FWM and AID, demonstrating its effectiveness.

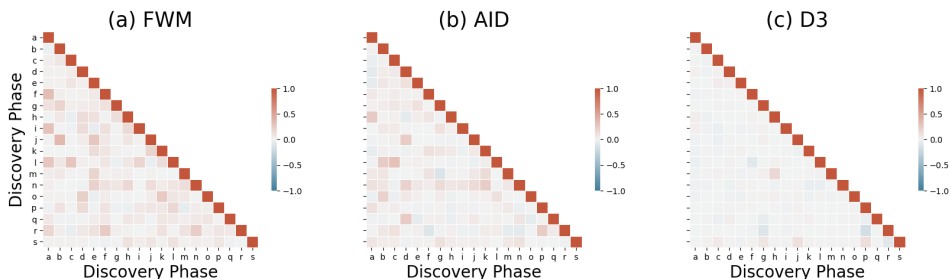

Figure 9: The heatmap displays the cosine similarity between the *roles* during the discovery phase for the SAR task.

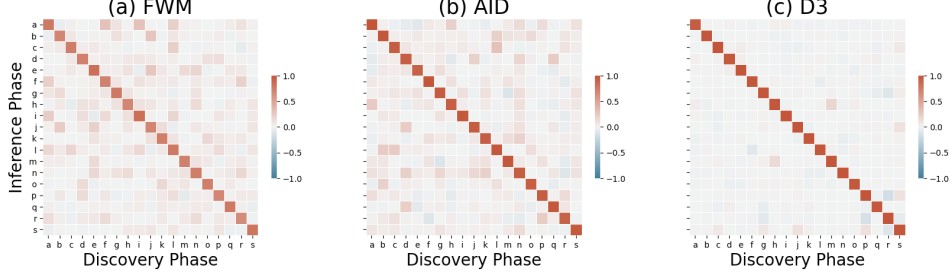

Figure 10: The heatmap displays the cosine similarity between the *roles* (**x-axis**) during the discovery phase and the *unbinding operators* (**y-axis**) during the inference phase for the SAR task.

## F.2 Qualitative Analysis for Different Seeds

Additionally, we present the results of the qualitative analysis for different seeds in the SAR task.

### F.2.1 $N_{\text{code}}$: 64, $D_{\text{code}}$: 32, top-$k$: 8, seed: 3333

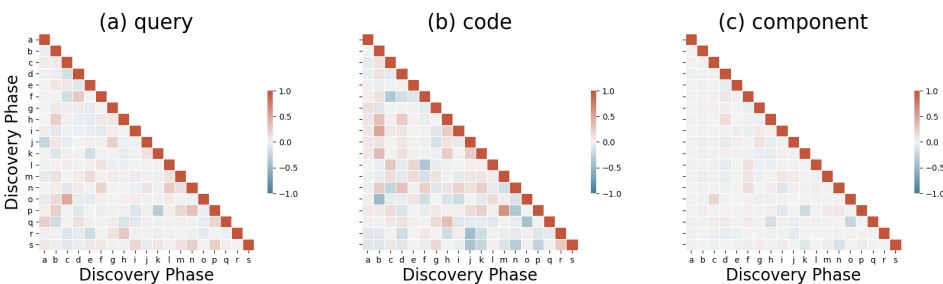

Figure 11: The heatmap displays the cosine similarity between the generated representations during the discovery phase for the SAR task. We explore the similarity across different types of representations: (a) `queries` of *roles*, (b) `codes` of *roles*, and (c) the *roles* themselves.

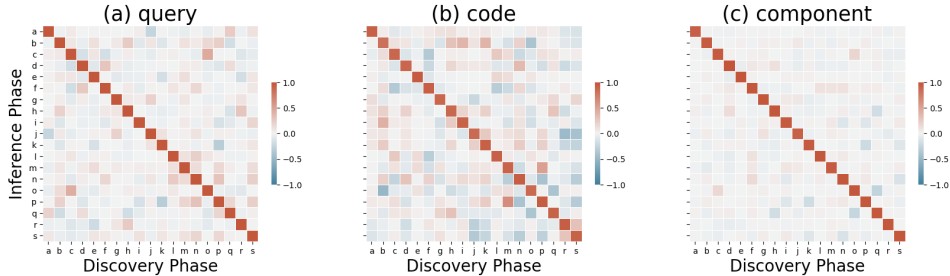

Figure 12: The heatmap displays the cosine similarity between the generated representations during the discovery phase (represented on the **x-axis**) and the inference phase (represented on the **y-axis**) for the SAR task. We explore the similarity across different types of representations: (a) `queries` of *roles* and *unbinding operators*, (b) `codes` of *roles* and *unbinding operators*, and (c) the *roles* and *unbinding operators* themselves.

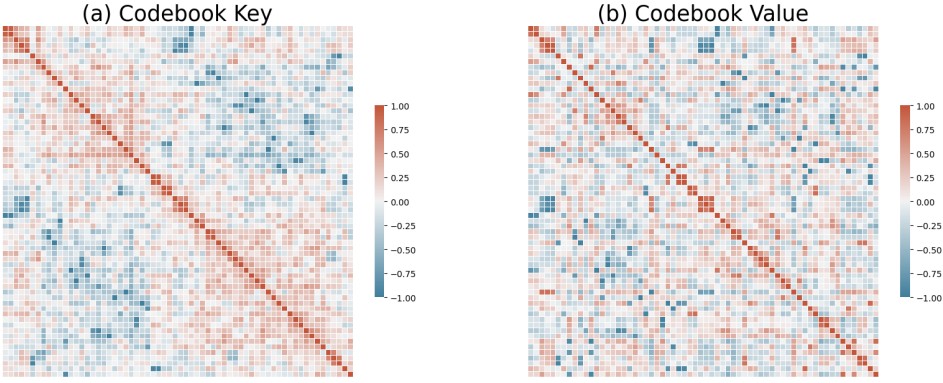

Figure 13: The heatmap visualizes the cosine similarity of the learned codebook features for the SAR task. There are two parts to each heatmap: (a) the similarity among codebook keys, denoted as $\{k_i\}_{i=1}^{N_{\text{code}}}$, and (b) the similarity among codebook values, denoted as $\{v_i\}_{i=1}^{N_{\text{code}}}$. For better visualization, the heatmap values are reordered to reflect the cluster of similar codebook keys.

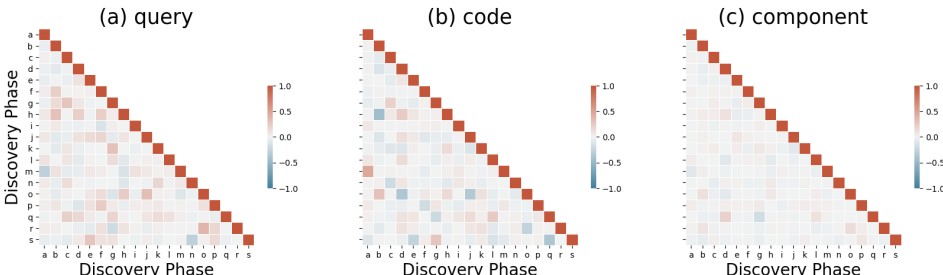

Figure 14: The heatmap displays the cosine similarity between the generated representations during the discovery phase for the SAR task. We explore the similarity across different types of representations: (a) `queries` of *roles*, (b) `codes` of *roles*, and (c) the *roles* themselves.

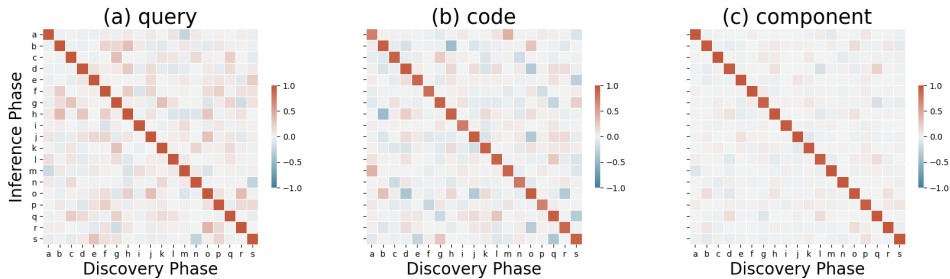

Figure 15: The heatmap displays the cosine similarity between the generated representations during the discovery phase (represented on the **x-axis**) and the inference phase (represented on the **y-axis**) for the SAR task. We explore the similarity across different types of representations: (a) `queries` of *roles* and *unbinding operators*, (b) `codes` of *roles* and *unbinding operators*, and (c) the *roles* and *unbinding operators* themselves.

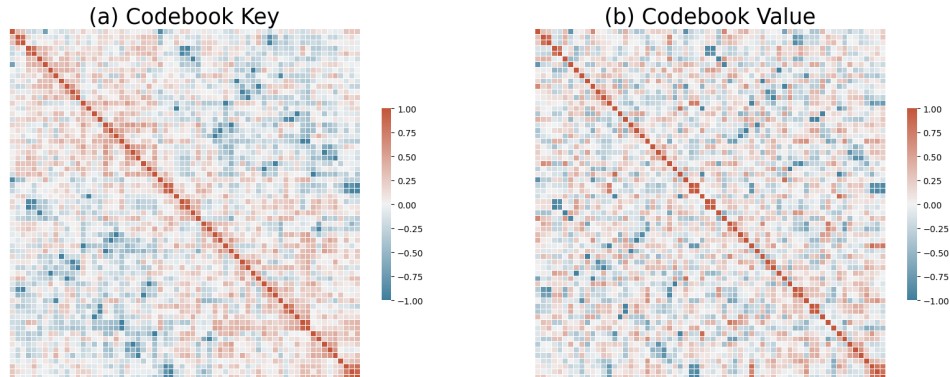

Figure 16: The heatmap visualizes the cosine similarity of the learned codebook features for the SAR task. There are two parts to each heatmap: (a) the similarity among codebook keys, denoted as $\{k_i\}_{i=1}^{N_{\text{code}}}$, and (b) the similarity among codebook values, denoted as $\{v_i\}_{i=1}^{N_{\text{code}}}$. For better visualization, the heatmap values are reordered to reflect the cluster of similar codebook keys.

