# OpenReview forum: "Discrete Dictionary-based Decomposition Layer for Structured Representation Learning"
_NeurIPS.cc/2024/Conference — NeurIPS 2024 poster_

### Official Review · Reviewer_FjyM · 2024-07-07

**Soundness:** 3
**Presentation:** 3
**Contribution:** 3
**Rating:** 6
**Confidence:** 4

**Summary:**

The authors propose the Discrete Dictionary-based Decomposition method, a discrete representation learning for TPRs. It encodes the roles and unbinding queries (and potentially also the fillers) using a learned dictionary, which serves as a codebook. The roles and then unbindings share the codebook, encouraging correct, noise-free retrieval. The layer is a drop-in replacement for any TPR-based layer. The authors demonstrate strong performance on tasks requiring systematic generalization, matching, or outperforming AID without using an iterative attention mechanism.

**Strengths:**

The authors present an interesting, novel, discrete representation of learning. Learning such representations is notoriously difficult. Their method is the only one that can solve the systematic split on the SAR task. The evaluation is solid. The authors have analyzed the learned roles and unbinding operators and showed that they are learning orthogonal representations without explicit regularization.

**Weaknesses:**

Given that there is a residual connection around the codebook, it is unclear what component of the roles and unbindings are generated by the codebook and which part of them comes from the residual. It would be nice to see ablations on the model without residual. If the model is not able to learn without it, an alternative is to replace the output of the residual by their mean over the dataset and see the effect on the performance.

Additionally, I would like to see analysis like Figures 3 and 4, but for the baselines without D3. Currently, in 282, the authors claim "indicating the effectiveness of D3", but it is unclear how orthogonal representations are the baselines' representations.

I believe that this additional analysis could deepen the understanding of why and how D3 works and significantly strengthen the paper.

**Questions:**

In line 115, the authors write: "This distinctive characteristic allows D3 to leverage the learned symbolic features to decompose unseen data after training.". Could the authors provide an additional explanation for why discretization aids the decomposition of unseen data, as opposed to denoising the representations?

On Fig 5 (b), the authors show an ablation over the number of keys in the codebook. Even with the smallest key size, the model works well. I would like to see a similar plot going down in the number of keys as long as the performance doesn't suffer significantly or the number of keys doesn't reach 1. This would help to understand better the effect of the codebook.

Calling WikiText 103 a large-scale language modeling dataset in 2024 is a stretch. Please just call it "language modeling" dataset.

**Limitations:**

The authors discuss the limitations of their method in the paper. As an additional limitation, as with all TPR papers, it is unclear how the improved generalization on toy tasks would transfer to real-world scenarios, like improving the generalization of LLMs. However, this should not be a reason to downvalue a contribution of the paper, given that it aims to improve an important shortcoming with a novel idea.

---

> ### Author Rebuttal · Authors · 2024-08-07
>
> We appreciate the reviewer for their constructive feedback and insightful suggestions. We will ensure that we reflect on our responses in our revised manuscript in the future.
>
> > W1: Ablation study for the effect of the residual connection
>
> **The *components* represent the vector representations of the symbolic components of TPR**, including roles, fillers, and unbinding operators, which are essential for TPR operations in TPR-based models. Specifically, D3 uses *queries* to access the codebook and generates *codes* based on the values in the codebook, as detailed in Eq. 3. Subsequently, it produces the *components* by residually connecting the *queries* to the *codes*, as described in Eq. 4. This mechanism implies that without the residual connection, the *components* are derived solely from the codebook values.
>
> In response to your suggestion, **we conducted an ablation study on the SAR and sys-bAbI tasks** to examine the impact of removing the residual connection. Please see the global response. **Fig. 13 shows that the residual connection is crucial for effectively training the D3 layer**.
>
> ***
>
> > W2: Additional orthogonal analysis on baseline models
>
> Thank you for your valuable input. In response, **we have conducted an orthogonal analysis for the baseline models** (FWM and AID) similar to the analysis presented in Section 4.3.1. Please see the global response. **Figs. 14 and 15 indicate that the D3 model generates more structured and orthogonal representations than the baseline models**, FWM and AID, demonstrating its effectiveness.
>
> ***
>
> > Q1: Could the authors provide an additional explanation for why discretization aids the decomposition of unseen data, as opposed to denoising the representations?
>
> Thank you for your insightful comment. Our work focuses on enhancing the systematic generalization of TPR-based approaches. Since systematic generalization aims at generalizing unseen data composed of known components observed during the training phase, the decomposition operations of TPR can be thought of as mapping unseen data to observed TPR components during training.
>
> The discretization technique enhances this decomposition operation by preserving the discrete features learned during training and enabling the mapping of given data to discrete representations. Within the D3 layer, **these discrete representations facilitate the generation of structured TPR representations that satisfy the TPR conditions**, as demonstrated in Figs. 3 and 4.
>
> Moreover, Figs. 3 and 4 illustrate that **the D3 layer also plays a role in denoising representations** by summing the *codes* to the *queries*, **effectively reducing the noise of the *queries* in the *components***. These characteristics allow the D3 layer to enhance the systematic generalization of TPR-based models by leveraging the discrete features to their decomposition operations of TPR.
>
> ***
>
> > Q2: Ablation study for the effect of varying the number of keys
>
> Thank you for your valuable input. In response to your comment, **we have expanded the range of our ablation study** to include additional experimental results for varying the number of keys in the codebook. Please see the global response. **Fig. 16 shows that even with a significantly reduced number of keys, the model with D3 maintains high accuracy on the SAR task**. This observation highlights the significance of the architectural inductive bias introduced by the D3 layer, effectively addressing the decomposition problem inherent in TPR-based models. These findings enhance our understanding of the codebook's impact on the model's performance.
>
> ***
>
> > Q3: The term "large-scale language modeling" in the manuscript
>
> **We will update the terminology to "language modeling"** to reflect its current standing better.
>
> ***
>
> > L1: As an additional limitation, as with all TPR papers, it is unclear how the improved generalization on toy tasks would transfer to real-world scenarios, like improving the generalization of LLMs.
>
> We appreciate the opportunity to address the concern regarding the transferability of improved generalization from toy tasks to real-world scenarios, a common limitation in TPR-related research.
> Previous work [1] has demonstrated the equivalence between TPR (or fast weight) and the linear attention mechanism, highlighting the potential for TPR to enhance linear attention. Additionally, recent works [2, 3] have proposed hybrid quadratic-linear attention methods that mitigate the computational complexity of self-attention while improving the generalization capabilities of LLMs.
>
> Given these findings, **we believe that the techniques developed in prior TPR-related research could address the limitations of linear attention within these hybrid methods, potentially improving the generalization of LLMs**. We plan to explore this promising direction in our future work.
>
> [1] Linear Transformers Are Secretly Fast Weight Programmers, ICML’21.
>
> [2] Transformer quality in linear time, ICML’22.
>
> [3] When Linear Attention Meets Autoregressive Decoding: Towards More Effective and Efficient Linearized Large Language Models, ICML’24

---

> > ### Comment · Reviewer_FjyM · 2024-08-10
> > **Concerns about the usefulness codebook**
> >
> > We are thankful to the authors for their efforts in the rebuttal, and they answered most of my questions positively.
> >
> > However, I find the results of "Q2: Ablation study for the effect of varying the number of keys *extremely concerning*. The performance of the network doesn't change with even as low as 2 keys in the codebook. This begs the question of whether the codebook, which is the main component of the model, is doing anything, or are there other differences in the training pipeline or the model that cause the observed difference in Fig 2 compared to the baselines? This makes me question the validity of the approach.
> >
> > Do the authors have an explanation of how this is possible? Can you please run additional ablations with (1) the codebook removed, but otherwise use the same code and pipeline, (2) the number of codebook elements set to 1, which is equivalent to having a linear layer down-projecting to 1 elements and then projecting it back.
> >
> > I would appreciate any evidence that shows that the codebook is crucial, like the one I described above, or one that show that codebooks with very small number of entries do not work on other tasks.

---

> > > ### Author Response · Authors · 2024-08-12
> > >
> > > Thank you for your detailed feedback and valuable suggestions.
> > >
> > > To address your concerns, **we conducted additional ablation studies on the SAR and sys-bAbI tasks**. Specifically, we investigated **the D3 layer without incorporating the codebook** (referred to as "*w/o codebook*"), where *components* are generated solely using the shared feed-forward networks (layer$\_\text{residual}$ and layer$\_\text{final}$). In this configuration, the *component* is calculated as "layer$\_\text{residual}$(*query*)" instead of the "*component* = *code* + layer$\_\text{residual}$(*query*)", which is described in Eq. 4.
> > >
> > > As shown in Table A1, even without the codebook, the D3 layer improves the generalization performance of the baseline model on the SAR task. This result indicates that **the shared feed-forward networks significantly contribute to performance enhancement on the SAR task**, which may explain why the model maintains robust performance even with fewer keys.
> > >
> > > However, it is important to note that without the codebook, the D3 layer does not achieve near-perfect accuracy on the SAR task (as shown in Table A1) and fails to significantly enhance the systematic generalization of the baseline on the sys-bAbI task (as shown in Table A2). **These results demonstrate that the codebook plays a crucial role in enhancing the model's overall performance and generalization capabilities**, especially in tasks requiring systematic generalization.
> > >
> > > Additionally, we experimented with $N_\text{code} = 1$ on the SAR task, where the codebook may act as a bias term. The results in Table A1 show that using a single codebook element results in degraded generalization performance compared to the "*w/o codebook*" configuration, indicating that multiple codebook elements are essential for achieving optimal results.
> > >
> > > | **Table A1.** SAR task | $D_\text{code}$ | $N_\text{code}$ | top-$k$ | Accuracy |
> > > | --- | :---: | :---: | :---: | :---: |
> > > | FWM | - | - | - | 44.9$_{\pm31.5}$ |
> > > | --- |
> > > | D3 | 32 | 4 | 2 | 87.38$_{\pm11.10}$ |
> > > | D3 | 32 | 64 | 8 | 99.27$_{\pm0.88}$ |
> > > | D3 *w/o codebook* | 32 | - | - | 89.02$_{\pm4.56}$ |
> > > | --- |
> > > | D3 | 64 | 1 | 1 | 89.10$_{\pm7.99}$ |
> > > | D3 | 64 | 4 | 2 | 94.47$_{\pm2.35}$ |
> > > | D3 | 64 | 64 | 8 | 94.29$_{\pm8.06}$ |
> > > | D3 *w/o codebook* | 64 | - | - | 91.65$_{\pm3.66}$ |
> > >
> > > | **Table A2.** sys-bAbI task | *w/o sys diff* | *w/ sys diff* |
> > > | --- | :---: | :---: |
> > > | FWM | 0.79$_{\pm0.14}$ | 2.85$_{\pm1.61}$ |
> > > | --- |
> > > | D3 | 0.75$_{\pm0.17}$ | 1.96$_{\pm0.88}$ |
> > > | D3 *w/o codebook* | 1.19$_{\pm0.41}$ | 3.55$_{\pm1.04}$ |
> > >
> > > In summary, **combining of the codebook and the shared residual networks is crucial for achieving systematic generalization in TPR-based models**. We hope these additional experiments provide clarity regarding our approach.

---

> > > > ### Comment · Reviewer_FjyM · 2024-08-12
> > > >
> > > > I appreciate the effort of the authors to clarify the effects of the codebook.
> > > >
> > > > The results show that the codebook is used, but the shared residual network may have a more significant impact. I think more analysis should be needed to determine the effect of each of these components. Additionally, more emphasis should be placed on the shared residual network, which is not discussed in the current version of the paper.
> > > >
> > > > Given this, I am maintaining my score.

---

> > > > > ### Author Response · Authors · 2024-08-13
> > > > >
> > > > > We sincerely appreciate Reviewer FjyM’s thorough review and valuable suggestions.
> > > > >
> > > > > In response to your feedback, **we will update our manuscript to include the additional experimental results you recommended and address your concerns** as follows:
> > > > >
> > > > > - Add ablation study for the effect of the residual connection (Figure 13)
> > > > > - Add additional orthogonal analysis on baseline models (Figures 14 and 15)
> > > > > - Add ablation study for the effect of varying the number of keys (Figure 16)
> > > > > - Add ablation study for the effect of the codebook (Tables A1 and A2)
> > > > >
> > > > > Furthermore, we will expand our discussion in the manuscript to emphasize the effect of both codebook and shared residual networks, which are the primary components of our D3 layer.
> > > > >
> > > > > The ablation study for the codebook on the SAR task (Table A1) indicates that the shared residual networks significantly enhance the generalization performance of the D3 layer. However, the results from the sys-bAbI task (Table A2) suggest that, while these networks improve performance, they alone struggle to generalize more structured data.
> > > > >
> > > > > The ablation studies for the codebook (Tables A1 and A2) demonstrate that incorporating the codebook mechanism results in nearly 100% accuracy on the SAR task and significantly improves the systematic generalization of models on the sys-bAbI task. However, as shown by the ablation study on the residual connection (Fig. 13), the codebook alone does not achieve the same level of generalization and exhibits instability in the D3 layer.
> > > > >
> > > > > In conclusion, our experimental results indicate that **the combination of the codebook and shared residual networks within the D3 layer is crucial for enhancing systematic generalization performance and stability**. By integrating these two components, **our D3 layer significantly enhances the systematic generalization capabilities of TPR-based models**.

---

> > > > > > ### Comment · Reviewer_FjyM · 2024-08-13
> > > > > >
> > > > > > I'm thankful for the author's effort to improve the paper. That makes the paper more correct, but I'm afraid I'm not convinced enough about the method to increase my score.
> > > > > >
> > > > > > Could the authors summarize what is the main difference between the shared residual (without a codebook) and the previous work?
> > > > > >
> > > > > > A suggestion if the authors want to keep working on improving the paper (not for the rebuttal, but for general curiosity): it might help to improve the codebook utilization and possibly the generalization properties if the authors would introduce a low-rank bottleneck on the residual, giving the network more incentive to rely on the codebooks.

---

> > > > > > > ### Author Response · Authors · 2024-08-14
> > > > > > >
> > > > > > > We sincerely thank the reviewer for thoughtful feedback and for acknowledging our efforts to improve the paper.
> > > > > > >
> > > > > > > The primary distinction in our approach lies in the application of shared residual networks. In the FWM, *queries* are treated as structured representations. In contrast, in the D3 layer, we introduced shared residual networks that are applied across all *queries*. This means that, regardless of the specific type of TPR components, the same residual network is employed to map *queries* into the dimensional space of *codes*. While FWM directly utilizes the *queries* as structured representations, our shared residual approach further transforms these *queries* through a shared feed-forward network before performing with subsequent TPR operations.
> > > > > > >
> > > > > > > We also appreciate the reviewer’s insightful suggestion. The suggested idea can potentially enhance the D3 layer by increasing the dependency on the codebook. We plan to explore this direction further in our future work.
> > > > > > >
> > > > > > > Thank you once again for your valuable input and your continued interest in our research.

---

### Official Review · Reviewer_3Nfz · 2024-07-12

**Soundness:** 2
**Presentation:** 2
**Contribution:** 2
**Rating:** 5
**Confidence:** 3

**Summary:**

The paper presents a discrete dictionary-based decomposition (D3) layer for tensor product representation (TPR). The purpose is to enhance the decomposition capabilities of TPR so that it can perform downstream tasks more effectively. D3 uses the discrete, trainable key-value dictionaries to map input data to “discrete” features within each dictionary. The dictionaries correspond to the TPR components, which are roles, fillers, and unbinding operators. More specifically, one dictionary is trained for the roles and unbinding operators and an optional one for the fillers. Experiments were conducted by using D3 in three TPR-based models. The paper shows that the use of D3 in these models outperforms the ones without using it.

**Strengths:**

-	The use of discrete dictionary-based method in TPR models seems novel. I am not working in the area. This novelty is claimed in the paper.
-	The related work section is well structured and clearly mentions the difference between the proposed method and existing ones.
-	The experiments were conducted with multiple TPR-based models on multiple tasks and the proposed method achieved the best results on most cases (except for one case in Table 1 in which FWM+AID is the best).

**Weaknesses:**

-	The paper assumes that the readers are familiar with terminologies used in the paper. It would be better if the paper provided definitions of some key concepts frequently used in the paper (such as roles, fillers, etc.). I had to look into references to fully understand the concepts.
-	D3 is evaluated with multiple TPR-based models on multiple tasks (such as text understanding and reasoning, visual relational reasoning and large language modeling). But it is unclear how these TPR-with-D3-based models compares with the SOTA models for these individual tasks. Without such a comparison, the significance of the work is unclear.
-	The paper mentioned at the beginning that using discrete representation can help with the interpretability of the models. However, there is no discussion or evaluation of this aspect of the model as the result of using D3.
-	I am not sure if the resulting “discrete” representation of the input data can be called “symbolic” features (which the paper uses a few times). The representations are still in the vector form, which is numeric. They may be considered “discrete”, but they are not symbols.
-	It is not clear how the dictionaries trained with D3 correspond to three components (roles, fillers, and unbinding operators) in TPR. Each component seems to be trained in the same way. There do not seem to be distinctions among them according to the descriptions in Section 3.1. In the experiments, only two or one dictionary is used, with one dictionary corresponding to roles and unbinding operators and an optional one for the fillers. If only the number of dictionaries matters, then the “w/o F” option could be considered “w/o roles/unbinding operators but with fillers”. More explanation on this would be beneficial.
-	There are some small errors in the presentation. For example, Subfigues (b) and (c) in Figure 6 should be switched to match their references in the text.

**Questions:**

-	How is the TPR-with-D3-based models compared with the SOTA method on their individual tasks?
-	Does the use of the D3 layer increase the interpretability of the models?
-	How do the dictionaries trained with D3 correspond to three components (roles, fillers, and unbinding operators) in TPR?

**Limitations:**

Yes.

---

> ### Author Rebuttal · Authors · 2024-08-07
>
> We appreciate the reviewer for their constructive feedback and insightful suggestions. We will ensure that we reflect on our responses in our revised manuscript in the future.
>
> > W1: More explanation for some key concepts
>
> We acknowledge that the initial description may have been challenging for readers unfamiliar with the TPR framework. We provide more detailed explanations of the key terms to enhance the comprehensibility of our paper as follows.
>
> The TPR is a general method for representing the symbolic structure of data using distributed representations. **This framework operates by explicitly decomposing data at the representation level into distinct symbols, such as *role-filler* pairs, which are then encoded through the tensor product of *role* vectors and *filler* vectors**, $T= filler \otimes\textit{role}$. By doing so, this encoding method preserves the symbolic structure of the data. The *roles and fillers* are dependent on the task at hand.
>
> For instance, in a tree structure, the *role* corresponds to the position within the tree, while the *filler* represents the associated label with that position [1]. In associative memory, the *role* is analogous to an associative key (or address), and the *filler* corresponds to the associative value (or contents) [2, 3].
>
> During the decoding phase, **the TPR framework retrieves specific *fillers*—essential for solving the given task—**from the encoded TPR representation via matrix multiplication using *unbinding operators* associated with particular *roles*, $filler = T \cdot \textit{unbind}$.
>
> In TPR-based neural networks, during training, **these TPR characteristics force the models to learn to generate structured representations satisfying TPR conditions through supervised training**. This process ensures the models can perform correct TPR operations to solve tasks accurately.
>
> [1] Differentiable tree operations promote compositional generalization, ICML’23.
>
> [2] TPR-RNN, NeurIPS’18.
>
> [3] FWM, ICLR’21.
>
> ***
>
> > W2 & Q1: Comparison with the SOTA models
>
> Our primary objective was to evaluate **how D3 enhances the systematic generalization capabilities of TPR-based models**. Therefore, **our initial experiments compared D3 with several TPR-based baselines across multiple tasks**. However, we acknowledge the importance of comparing our D3-enhanced TPR models with SOTA methods for a comprehensive evaluation.
>
> In response to your inquiry, we have expanded our comparisons to include a broader range of state-of-the-art methods. Please see the global response.
>
> ***
>
> > W3 & Q2: Interpretability of D3
>
> The TPR framework decomposes data at the representation level into distinct symbols, such as *role-filler* pairs for encoding and unbinding operators for decoding. **This characteristic of TPR improves the interpretability of models** because the relationship between *the roles and the unbinding operators* explains which parts of the input the model focuses on to predict the answer. However, **this interpretability is accurate only when the generated structured representations satisfy the TPR conditions**.
>
> In this context, **the D3 layer enhances the interpretability of the models by providing structured representations that more effectively satisfy the TPR conditions compared to baseline models**. In response to Reviewer FjyM, we have included experimental results of the orthogonal analysis for the baseline models. Please see the global response. Figs. 14 and 15 demonstrate that the generated representations by the D3 better confirm the TPR conditions than other baseline models, supporting our claim that the D3 layer contributes to increased interpretability.
>
> ***
>
> > W4: The term "symbolic" features in the manuscript
>
> Thank you for your valuable input. Although we intended to convey that the codebook features within the dictionaries capture information that could be associated with symbolic components, it is more accurate to refer to these features as "discrete" rather than "symbolic" since they remain in vector form and are numeric. **We will revise the term "symbolic feature" to "discrete feature"** to prevent any potential misunderstanding (specifically on lines 13, 60, 61, 71, and 116).
>
> ***
>
> > W5 & Q3: How do the dictionaries trained with D3 correspond to TPR components?
>
> We appreciate the opportunity to clarify the correspondence between the dictionaries trained with D3 and the TPR components in TPR.
>
> As discussed in response to W1, during the training phase, the decomposition module in TPR-based models learns to generate TPR components to perform TPR operations. **Each dictionary in D3 is explicitly linked to a specific TPR component, ensuring that each dictionary is responsible only for generating its corresponding component**. The generated components are then utilized in pre-defined TPR operations of the TPR-based models. **This setup ensures that each dictionary is trained to be specialized to a specific TPR component**.
>
> Moreover, the number of dictionaries directly correlates with the TPR operations of the baseline models. For instance, the TPR operations of the FWM require two types of roles and one filler for encoding ($T= filler \otimes\textit{role}_1 \otimes \textit{role}_2$) and two types of unbinding operators for decoding ($filler = T \cdot \textit{unbind}_2 \cdot \textit{unbind}_1$). We thus set up three distinct dictionaries when integrating D3 with FWM (one for $filler$, another for $role_1/unbind_1$, and the other for $role_2/unbind_2$).
>
> We opted for the configuration "*with roles/unbinding operators*" to allow the D3 layer to contribute to generating structured representations that satisfy the TPR conditions, regardless of the number of dictionaries. Your suggested option, "*w/o roles/unbinding operators but with fillers*," presents an interesting alternative for examining the design of D3 and could be considered for future studies.
>
> ***
>
> > W6: Small errors in Figs. 6(b) and (c)
>
> **We will correct the captions**.

---

> > ### Comment · Reviewer_3Nfz · 2024-08-14
> >
> > Thank you for the clarification. I've adjusted my rating accordingly.

---

> > > ### Author Response · Authors · 2024-08-14
> > >
> > > We thank Reviewer 3Nfz for your time and your constructive comments during the rebuttal period.

---

### Official Review · Reviewer_QJJM · 2024-07-12

**Soundness:** 2
**Presentation:** 3
**Contribution:** 3
**Rating:** 6
**Confidence:** 3

**Summary:**

Drawing inspiration from discrete representation learning with dictionaries, the author introduced a novel Tensor Product Representation (TPR) framework, a Discrete Dictionary-based Decomposition (D3) layer designed to retain the learned symbolic features during training and apply them effectively to address decomposition challenges in previously unseen data. Owing to its architectural properties, this method is defined as a drop-in layer that maps input to pre-learned symbolic features, facilitating smooth integration with existing TPR methods. Comprehensive experimental results of this novel approach showcase superior or comparable performance to that of other baseline methods.

**Strengths:**

- S1: Generating components in the proposed method is analogous to transformer blocks with Top-K selection, which is straightforward and intuitive.
- S2: Experimental results and ablation studies are well-organized and highlight the effectiveness of the proposed method.

**Weaknesses:**

- W1: The proposed method's technical novelty is a slight improvement over the previous work, AID. It appears that the proposed method's performance in more complex cases, such as Sort-of-CLEVR and WiKiText, is due to the introduction of additional configuration complexity and more learnable parameters. This may cause scalability issues in cases where more complex TRP operations are required since the number of dictionaries a model directly corresponds to the number of roles/unbind operators.
- W2: Although the proposed method is a drop-in layer applicable to other existing TPR methods, it raises the question of how such an attention key-query-value mechanism can meet the TPR operation conditions. In particular, how can you ensure that the shared dictionary satisfies these properties during training to satisfy near orthogonality between roles or between unbinding operators?

**Questions:**

Please check out the Weakness section first. I listed additional questions as follows:
- Q1: In Fig 2, the experimental results of the SAR task appear to show a huge difference in performance between the proposed method and AID. In the original AID paper, it achieved about 90% accuracy for the SAR task. The original experiment setup that AID ran on appears to be more difficult than yours, so why did AID perform significantly worse on your setup?
- Q2 (related to W2):  In Fig. 5(a), there are some strong similarities among specific keys, which indicates some redundancy of the learned keys. How can these redundant properties of keys not violate the TPR operation conditions?

**Limitations:**

Please check out the Weakness section.

---

> ### Author Rebuttal · Authors · 2024-08-07
>
> We appreciate the reviewer for their constructive feedback. We will ensure that we reflect on our responses in our revised manuscript in the future.
>
> > W1: Technical novelty over AID and scalability issue
>
> While the prior work introduced an iterative competitive attention-based decomposition module, called AID, which refines structured representations iteratively, it has limitations in generalizing to unseen compositions of known symbols even in simple synthetic tasks, as shown in Fig. 2. AID lacks an explicit mechanism to leverage observed structural information during training for decomposition of TPR.
>
> To address these limitations, **we propose a novel discrete representation-based decomposition method for TPR-based models, which stores discrete features during training and maps given data to these learned discrete features via sparse activation**.
>
> Our experimental results in Tables 2 and 3 demonstrate that D3 achieves comparable performance to AID with a similar parameter setting ($D_\text{code}$ of 128 on the CLEVR and 32 on the Wiki). Moreover, D3 performs better with increased parameters (256 on the CLEVR task and 64 on the Wiki). **These results highlight our method's technical novelty and effectiveness compared to AID**.
>
> The scalability of our approach is inherently linked to TPR operations of baseline models since the number of dictionaries in the D3 layer aligns with the number of TPR components required for their operations. As TPR operations require increasing components to handle large datasets, our method also requires a proportional increase in dictionaries, resulting in significant computational and memory overhead. As explored in prior work, one potential solution to mitigate this issue is distributing shared dictionaries across multiple heads or layers [1]. However, this approach requires further investigation and experimentation, which we plan to research in future work.
>
> [1] Large memory layers with product keys, NeurIPS’19.
>
> ***
>
> > W2: How can you ensure that the shared dictionary satisfies these properties during training to satisfy near orthogonality between roles or between unbinding operators?
>
> The TPR framework operates by explicitly decomposing data at the representation level into distinct symbols, such as *role-filler* pairs, which are then encoded through the tensor product of *role* vectors and *filler* vectors, $T= filler \otimes\textit{role}$. By doing so, this encoding method preserves the symbolic structure of the data. During decoding, the framework retrieves specific *fillers*—essential for solving the given task—via matrix multiplication using *unbinding operators* associated with particular *roles*, $filler = T \cdot \textit{unbind}$. These TPR characteristics force TPR-based models to learn to generate structured representations satisfying TPR conditions through supervised training.
>
> **When integrating with TPR-based models, our D3 layer is trained to produce structured representations that satisfy these TPR conditions**. D3 employs a sparse key access mechanism during the generation of these representations. This mechanism ensures that individual discrete features within dictionaries become specialized to specific latent data features. Consequently, this specialization allows each discrete feature to learn and indirectly satisfy the near orthogonality requirements of TPR properties, thereby maintaining accurate TPR operations.
>
> ***
>
> > Q1: Difference in experimental settings of the SAR task between our work and the AID paper
>
> In the original AID paper, the SAR task performance was evaluated with varying levels of the task parameter $p$ with different values (0.0, 0.5, and 1.0), which adjusts the combinations of symbols observed during training. The AID achieved about 90% accuracy for $p$ values of 0.5 and 1.0, which are less challenging settings.
>
> **Our experiment adopts the SAR task's most challenging setting ($p=0.0$)**. This setting results in a more rigid evaluation environment than the other configurations. Consequently, the AID method does not achieve the same level of performance in our setup as it did in the original paper.
>
> To ensure clarity and prevent potential misinterpretation, we will include a detailed description of these differences between our work and the AID paper in the revised manuscript. This will help readers understand the context and rationale behind the observed performance disparities.
>
> ***
>
> > Q2: How can these redundant properties of keys not violate the TPR operation conditions?
>
> The D3 layer generates structured TPR representations by mapping input data to pre-learned discrete features within dictionaries through three steps: (1) *query* generation, (2) sparse key access, and (3) aggregation of code values. Therefore, the keys within dictionaries are intermediate features that assist the D3 layer in generating structured representations that satisfy the TPR conditions rather than needing to satisfy those conditions themselves.
>
> While some codebook keys in Fig. 5(a) show strong similarities, their corresponding codebook values in Fig. 5(b) show near-orthogonal patterns. This implies that **when similar *queries* access the dictionaries via these codebook keys, the resultant codes (read from the dictionaries) are orthogonal**. Figs. 3 and 4 provide further evidence supporting this claim.
>
> **As illustrated in Figs. 3 and 4, even though the *queries* are similar, the *codes* (retrieved from dictionaries via the *queries*) show more orthogonal patterns than the *queries* themselves**. Additionally, Figs. 3(c) and 4(c) demonstrate that D3 successfully generates *components* (which are our main focus) that meet the TPR conditions using these intermediate features (*queries and codes*).
>
> We believe this comprehensive approach ensures that the redundant properties of the keys do not violate TPR operation conditions.

---

> > ### Comment · Reviewer_QJJM · 2024-08-12
> > **Response to the reviewers' rebuttal**
> >
> > Thank you for the authors' clarification and for addressing my concerns.
> > I have decided to increase the score.

---

> > > ### Author Response · Authors · 2024-08-13
> > >
> > > We thank Reviewer QJJM for your time and your constructive comments during the rebuttal period.

---

### Official Review · Reviewer_KCxS · 2024-07-13

**Soundness:** 3
**Presentation:** 3
**Contribution:** 3
**Rating:** 6
**Confidence:** 4

**Summary:**

This paper address the difficulty of decomposing input data into Tensor Product Representation (TPR) components, namely, roles, fillers, and unbinding operators. The proposal called D3 includes the use of learnable dictionaries for these components, and the mapping of input data into intermediate features (code, query, component), in order to generate TPR components from input data. The authors demonstrate that D3 can be easily integrated into existing TPR-based models and improves their systematic generalization performance across various tasks, including synthetic associative recall, text/visual question-answering, and language modeling.

**Strengths:**

* Originality: The D3 method presents a novel approach to addressing the decomposition problem in TPR-based models. The use of discrete, learnable dictionaries for this purpose is innovative.

* Quality: The paper demonstrates thorough experimentation across multiple tasks and provides detailed ablation studies to support its claims.

* Clarity: The method is explained clearly, with helpful visualizations and step-by-step descriptions of the D3 layer.

* Significance: Improving the decomposition capabilities of TPR-based models has potential implications for enhancing systematic generalization in neural networks, which is an important goal in AI research.

**Weaknesses:**

* The paper is an alternative to the recently introduced AID method (Ref #22), and the goal/settings are nearly identical. It is helpful to clearly differentiate the contributions of this work to AID.
* There are design choices in the D3: dictionaries, query, code, components and the ways they are connected. There should be clear motivations for each of these choice.
* While the paper provides some analysis of the learned representations, a deeper theoretical analysis of why D3 works well could enhance the contribution.
* The experiments are primarily focused on synthetic or relatively simple tasks. Testing on more complex, real-world tasks would strengthen the paper's claims about generalization. The comparison to baselines could be expanded to include a wider range of state-of-the-art methods in compositional generalization.

**Questions:**

The questions naturally arise from the weaknesses above.

**Limitations:**

The authors discuss some limitations of their approach, including the need for task-specific configuration and added computational overhead. They could expand on potential limitations in terms of scalability to very large models/datasets.

---

> ### Author Rebuttal · Authors · 2024-08-07
>
> We appreciate the reviewer for their constructive feedback and insightful suggestions. We will ensure that we reflect on our responses in our revised manuscript in the future.
>
> > W1: Comparison to AID
>
> While our work and AID focus on enhancing the systematic generalization of TPR-based models by addressing a decomposition problem, our contributions differ from AID's in several important aspects. Our main contribution is **introducing a novel decomposition method for the decomposition of TPR**, which stores discrete features during training and maps given data to these learned discrete features via sparse activation.
>
> In detail, AID employs a competitive attention mechanism between input features and intermediate TPR representations, refining these representations iteratively. However, as shown in Fig. 2, AID has difficulties generalizing to unseen compositions of known symbols. Additionally, AID does not explicitly leverage observed structural information during training. To address these limitations, we propose the D3 layer, which maps input data to the nearest pre-learned discrete features within dictionaries, generating structured TPR representations. Our comprehensive experiments demonstrate that the D3 layer significantly improves the systematic generalization of TPR-based models.
> ***
>
> > W2: Design choices in the D3
>
> Our work focuses on enhancing the systematic generalization of TPR-based models. Since systematic generalization aims at generalizing unseen data composed of known components observed during training, the decomposition of TPR can be thought of as mapping input data to observed TPR features during training. This insight inspired our design choices for the D3 layer.
>
> **Our primary design choice was to employ a discrete representation-based decomposition module, which maps input data to discrete features learned during training**. Inspired by a prior key-value architecture [1], we introduced separate key-value-based dictionaries, each linked to specific TPR components (*role, filler, and unbinding operator*). This design ensures that each dictionary captures the symbolic information of the specific TPR components. The D3 layer generates *queries* based on the input data, which are then used to search and identify discrete features associated with the input data. Once relevant discrete features are identified, the codebook values are aggregated to generate *codes*. These *codes and queries* are then utilized as components for TPR operations to solve downstream tasks.
>
> In summary, each design choice in the D3 layer—from the multiple discrete dictionaries to the generation and utilization of *queries and codes*—was motivated by **the goal of mapping input data to discrete, learned features that facilitate systematic generalization in the decomposition operation of TPR**.
>
> [1] Large memory layers with product keys, NeurIPS’19.
>
> ***
>
> > W3: A deeper theoretical analysis
>
> We appreciate the suggestion to provide a deeper theoretical analysis of why D3 works well.
>
> Our D3 employs a separate key-value-based discretization mechanism, the robustness of which against distributional shifts was theoretically investigated in prior work [2]. **The D3 layer enables TPR-based models to mitigate errors in generating structured representations by mapping input data to pre-learned discrete features**. Figs. 3 and 4 empirically demonstrate how D3 mitigates errors while generating structured representations. Initially, generated *queries* fail to satisfy the TPR conditions, potentially causing inaccuracies in TPR operations. D3 addresses this by leveraging the *codes* derived through the discretization and generates near-ideal structured representations. These results imply that D3 improves the robustness of TPR-based models to unseen data comprising known symbols, thus enhancing the model's overall performance.
>
> Furthermore, in response to Reviewer FjyM, we have included further orthogonal analysis for the baseline models. Please see the global response. Figs. 14 and 15 show the superior quality of representations produced by D3 compared to the baseline models, further demonstrating the effectiveness of D3.
>
> [2] Discrete key-value bottleneck, ICML’23
>
> ***
>
> > W4: Additional experimental comparison
>
> Our experiments evaluated the enhancement of systematic generalization in TPR-based models achieved by D3. We compared D3 with several TPR-based baselines across various systematic generalization tasks. Additionally, to assess the effectiveness of D3 in more complex scenarios, we extended our evaluation to the WikiText-103 task. **While those tasks considered in our study are relatively simple, the results consistently demonstrate that D3 significantly improves the generalization performance of TPR-based models**. We acknowledge the comment regarding the need for testing on more complex, real-world tasks. Although our current evaluation provides strong evidence of D3's benefits in systematic generalization tasks, further investigation is needed to confirm its effectiveness in real-world scenarios. We plan to address this limitation in future work.
>
> In response to your suggestion, we have expanded our comparisons to include a broader range of state-of-the-art methods. Please see the global response.
>
> ***
>
> > L1: Scalability of D3
>
> The scalability of our approach is inherently linked to TPR operations of baseline models since the number of dictionaries in the D3 layer aligns with the number of TPR components required for their operations. As TPR operations require increasing components to handle large datasets, our method also requires a proportional increase in dictionaries, resulting in significant computational and memory overhead. As explored in prior work, one potential solution to mitigate this issue is distributing shared dictionaries across multiple heads or layers [1]. However, this approach requires further investigation and experimentation, which we plan to research in future work.

---

> > ### Comment · Reviewer_KCxS · 2024-08-14
> >
> > Thank you for detailed response!

---

### Author Rebuttal · Authors · 2024-08-07

## Global response

We thank all the reviewers for their constructive feedback and insightful suggestions. We believe that the additional ablation studies and comparisons they recommended provide a clearer understanding of D3's strengths and limitations. Our revised manuscript will thoroughly reflect these improvements.

We have attached a one-page PDF that includes the additional ablation studies suggested by reviewer FjyM, which are as follows:
- Ablation study on the effect of the residual connection (Figure 13)
- Additional orthogonal analysis on baseline models (Figures 14 and 15)
- Ablation study on the effect of varying the number of keys (Figure 16)

Moreover, as reviewers KCxS and 3Nfz suggested, we have expanded our comparisons to include a broader range of state-of-the-art methods, as detailed below.

**sys-bAbI task**: We compared D3 to state-of-the-art methods (DAM [1] and STM [2]) on the original bAbI task. The results in the table below show that existing memory networks struggle with the sys-bAbI task, highlighting the efficacy of D3 compared to these state-of-the-art memory networks.

| **sys-bAbI task** | *w/o sys diff* | *w/ sys diff* | Gap |
| --- | :---: | :---: | :---: |
| TPR-RNN | 0.79$_{\pm0.16}$ | 8.74$_{\pm3.74}$ | 7.95 |
| TPR-RNN (+AID) | 0.69$_{\pm0.08}$ | 5.61$_{\pm1.78}$ | 4.92 |
| TPR-RNN (+D3) | 0.65$_{\pm0.25}$ | 3.50$_{\pm2.07}$ | 2.85 |
| FWM | 0.79$_{\pm0.14}$ | 2.85$_{\pm1.61}$ | 2.06 |
| FWM (+AID) | 0.45$_{\pm0.16}$ | 1.21$_{\pm0.66}$ | 0.76 |
| FWM (+D3 *w/ F*) | 0.75$_{\pm0.17}$ | 1.96$_{\pm0.88}$ | 1.21 |
| --- |
| DAM [1] | 0.48$_{\pm0.20}$ | 5.25$_{\pm1.64}$ | 4.77 |
| STM [2] | 0.49$_{\pm0.16}$ | 4.19$_{\pm1.53}$ | 3.7 |

**Sort-of-CLEVR task**: We included the Compositional Transformer [3], designed to enhance the systematic generalization capabilities of multi-head self-attention methods. (our experiment was performed under identical experimental settings as this prior work [3].) The results show that the Linear Transformer significantly degrades systematic generalization performance compared to the Transformer. While D3 improves the performance of the Linear Transformer from a TPR perspective, it still shows limited performance in reasoning the relationships between multiple objects (*Binary* and *Ternary*) compared to the Transformer and Compositional Transformer.

| **Sort-of-CLEVR task** | $D_\text{code}$ | *Unary* | *Binary* | *Ternary* |
| --- | :---: | :---: | :---: | :---: |
| Linear Transformer | - | 69.3$_{\pm14.8}$ | 75.5$_{\pm1.3}$ | 56.4$_{\pm4.3}$ |
| Linear Transformer (+AID) | - | 98.9$_{\pm0.2}$ | 78.6$_{\pm0.3}$ | 63.7$_{\pm1.2}$ |
| Linear Transformer (+D3 *w/ F*)| 128 | 98.9$_{\pm0.2}$ | 79.5$_{\pm0.8}$ | 63.1$_{\pm1.9}$ |
| Linear Transformer (+D3 *w/ F*) | 256 | 99.0$_{\pm0.3}$ | 82.1$_{\pm2.4}$ | 68.8$_{\pm1.2}$ |
| --- |
| Transformer | - | 97.4$_{\pm3.5}$ | 84.3$_{\pm4.3}$ | 62.7$_{\pm3.9}$ |
| Compositional Transformer [3] | - | 98.9$_{\pm0.2}$ | 88.4$_{\pm1.4}$ | 66.5$_{\pm1.9}$ |


**WikiText-103 task**: We compared D3 to the Delta Network [4], which introduced a delta updating rule instead of the additive outer product-based updating rule in the Linear Transformer. (our experiment was performed under identical experimental settings as this prior work [4].) The results indicate that although D3 improves the performance of the Linear Transformer in language modeling tasks, the choice of updating rules has a more substantial impact on performance for tasks involving the comprehension of lengthy corpora than the decomposition operation.

| **WikiText-103 task** | $D_\text{code}$ | Valid | Test |
| --- | :---: | :---: | :---: |
| Linear Transfomer | - | 36.473 | 37.533 |
| Linear Transfomer (+AID) | - | 36.159 | 37.151 |
| Linear Transfomer (+D3 *w/o F*) | 32 | 36.061 | 37.220 |
| Linear Transfomer (+D3 *w/o F*) | 64 | 35.975 | 37.009 |
| --- |
| Delta Network [4] | - | 35.640 | 36.659 |

[1] Distributed associative memory network with memory refreshing loss. *Neural Networks*, *144*, 33-48.

[2] Self-attentive associative memory, ICML’20.

[3] Compositional Attention: Disentangling Search and Retrieval, ICLR’22.

[4] Linear transformers are secretly fast weight programmers, ICML’21.

---

### Decision · Program_Chairs · 2024-09-25

**Decision:**

Accept (poster)

**Comment:**

The paper proposes D3, a layer of discrete, learnable dictionaries for the Tensor Product Representation (TPR) framework. With each dictionary connected to a specific TPR component (e.g., roles, fillers, unbinding operators), D3 enables an effective decomposition of input data into TPR components. Experiments with 3 different TPR architectures on 4 different tasks demonstrate the benefits of the proposed layer compared to baselines without it and alternative solutions to the decomposition problem in TPR architectures.

All 4 reviewers feel cautiously positive about the paper, commending its novel approach to the decomposition problem in TPR architectures, thorough evaluation, and insightful analyses. The main concerns expressed by the reviewers were related to (1) architectural / performance differences relative to prior work — primarily AID — with similar goals; (2) missing ablation studies / comparisons to state-of-the-art baselines; (3) the relationship between the dictionaries and the TPR components, in particular in the context of interpretability and enforcement of TPR conditions; and (4) scalability. Accessibility was noted as a potential issue for readers without familiarity with the TPR framework.

The authors responded well to these concerns. They convincingly positioned D3 relative to AID (both in terms of technical novelty and practical performance), provided the requested ablation studies and comparisons to state-of-the-art baselines, and discussed D3 design choices and their effects on TPR conditions / interpretability. It is expected that these discussions and results, as well as a gentle introduction to the TPR framework, will be integrated into the paper. Scalability was identified as an issue that requires further attention.

Given that the major concerns have been addressed, as well as the relevance of the problem and novelty of the proposed approach, the paper is recommended for acceptance as a poster.